# Decision-Theoretic Approaches for Improved Learning-Augmented Algorithms

**Spyros Angelopoulos**
CNRS and International Laboratory
on Learning Systems
Montreal, Canada

**Christoph Dürr**
Sorbonne University, CNRS, LIP6
Paris, France

**Georgii Melidi**
Sorbonne University, CNRS, LIP6
Paris, France

## Abstract

We initiate the systematic study of decision-theoretic metrics in the design and analysis of algorithms with machine-learned predictions. We introduce approaches based on both deterministic measures such as *distance*-based evaluation, that help us quantify how close the algorithm is to an ideal solution, and stochastic measures that balance the trade-off between the algorithm's performance and the *risk* associated with the imperfect oracle. These techniques allow us to quantify the algorithm's performance across the full spectrum of the prediction error, and thus choose the *best* algorithm within an entire class of otherwise incomparable ones. We apply our framework to three well-known problems from online decision making, namely ski rental, one-max search, and contract scheduling.

## 1 Introduction

The field of learning-augmented computation has experienced remarkable growth recently. The focus, in this area, is on algorithms that leverage a machine-learned *prediction* on some key elements of the input, based on historical data. The objective is to obtain algorithms that outperform the pessimistic, worst-case guarantees that apply in the standard settings. Online algorithms with ML predictions were first studied systematically in Lykouris & Vassilvtiskii (2018) and Purohit et al. (2018a) and since then, the learning-augmented lens has been applied to numerous settings, including rent-or-buy problems (Gollapudi & Panigrahi, 2019), graph optimization (Azar et al., 2022), secretaries (Antoniadis et al., 2023), packing and covering (Bamas et al., 2020), and scheduling (Lattanzi et al., 2020). This is only a representative list; see the repository (Lindermayr & Megow, 2025).

A major challenge in learning-augmented algorithms is the theoretical analysis, and its interplay with the design considerations. Unlike the standard model, which focuses on the performance on worst-case inputs such as the *competitive ratio* (Borodin & El-Yaniv, 1998), the analysis of algorithms with predictions is multi-faceted, and involves objectives in trade-off relations. Typical desiderata require that the algorithm has good *consistency* (informally, its performance assuming a perfect, error-free prediction) as well as *robustness* (i.e., its performance under an arbitrarily bad prediction of unbounded error). Beyond these two extremes, there is an additional natural requirement that the algorithm's performance degrades *smoothly* as a function of the prediction error.

It is unsurprising that not all of the above objectives can always be attained and simultaneously optimized (Lavastida et al., 2021). Such inherent analysis limitations have an important effect on the algorithm's design. One concrete methodology is to design algorithms that optimize the trade-off between consistency and robustness, often called *Pareto-optimal* algorithms; e.g. (Sun et al., 2021a; Lee et al., 2024; Wei & Zhang, 2020; Christianson et al., 2023). Another design approach is to enforce smoothness, without quantifying explicitly the loss in terms of consistency or robustness, e.g., (Angelopoulos et al., 2022; Antoniadis et al., 2023).

Each approach has its own merits, but also certain deficiencies. Pareto-optimality may lead to algorithms that are *brittle*, in that their performance may degrade dramatically even in the presence of imperceptible prediction error (Elenter et al., 2024). From a practical standpoint, this drawback renders such algorithms highly inefficient. Even if brittleness can be avoided, one may obtain an entire *class* of Pareto-optimal algorithms, whose members exhibit incomparable smoothness (Benomar & Perchet, 2025). On the other hand, smoothness can often be enforced by assuming an upper bound on the prediction error, which can be considered, informally, as the *confidence* in the prediction oracle or the *tolerance* to prediction errors. The design and the analysis are then both centered around this confidence parameter (Angelopoulos et al., 2022; Antoniadis et al., 2023). However, this approach leads to algorithms that may be inferior for a large range of the prediction error, and notably when the prediction is highly accurate (i.e., the error is small). It also requires either explicit knowledge or an ad-hoc choice of this confidence value.

We are thus confronted with the following central question: *Among the many possible algorithms, each with its own performance function, how to choose the "best"?* Here, the performance function is the smoothness that interpolates between the extreme points of consistency and robustness. Answering this question hinges on the choice of a principled measure for the comparison of performance curves, which is typically the purview of decision theory and the focus of this work.

**Three classic problems: ski rental, one-max search, and contract scheduling.** To demonstrate our framework, we consider three classic problems. Our first problem, namely *ski rental*, is a classic formulation of rent-or-buy settings, and has served as proving grounds for learning-based algorithmic approaches. Given an unknown horizon of days, the decision-maker must decide on which day to stop renting, and irrevocably buy the equipment. The best deterministic competitive ratio is 2 (Karlin et al., 1988) (assuming a continuous setting), however a prediction on the horizon length can help improve the competitive ratio, as has been shown in several works on this problem and its extensions (Angelopoulos et al., 2020; Purohit et al., 2018b; Wei & Zhang, 2020; Khanafer et al., 2013; Gollapudi & Panigrahi, 2019; Wang et al., 2020a; Zhao et al., 2024). Pareto-optimal algorithms were studied in Wei & Zhang (2020); Angelopoulos et al. (2020); Purohit et al. (2018a). Furthermore, the recent work of Benomar & Perchet (2025) described a parameterized class of algorithms, all of which are Pareto-optimal, but exhibit different, and incomparable smoothness.

A second problem that is fundamental in sequential decision making is *one-max search*, in which a trader aims to sell an indivisible asset. Here, the input is a sequence $\sigma$ of *prices*, and the trader must accept one of the prices in $\sigma$ irrevocably. The problem and its generalizations have a long history of study, see e.g. El-Yaniv et al. (2001); Mohr et al. (2014); Clemente et al. (2016); Damaschke et al. (2009); Lee et al. (2024) as well as Chapter 14 in Borodin & El-Yaniv (1998). The learning-augmented setting in which the algorithm leverages a prediction on the maximum price in $\sigma$ was studied in Sun et al. (2021a), which gave Pareto-optimal algorithms. However, this algorithm suffers from brittleness (Elenter et al., 2024). Angelopoulos et al. (2022) gave an algorithm with smooth error degradation, but no consistency/robustness guarantees, based on a tolerance parameter $\delta$. However, this algorithm has inferior performance if the prediction is highly accurate.

Last, we consider a problem that is fundamental in real-time systems and bounded-resource reasoning in AI, namely *contract scheduling* (Russell & Zilberstein, 1991; Bernstein et al., 2003; López-Ortiz et al., 2014). Here, the aim is to design a system with interruptible capabilities via executions of a non-interruptible algorithm. The performance of the system is measured by the *acceleration ratio*, i.e., the multiplicative loss due to the repeated executions. Angelopoulos & Kamali (2023b) studied the setting in which an oracle predicts the interruption time, and gave a Pareto-optimal schedule. However, all Pareto-optimal algorithms are brittle, as shown in Elenter et al. (2024). Assuming a tolerance $\delta$ on the range of the prediction error, Angelopoulos & Kamali (2023b) also gave a schedule that has better smoothness, but is once again inefficient for small prediction error.

## 1.1 CONTRIBUTIONS

We present the first principled study of decision-theoretic approaches in learning augmented algorithms. Our objective is to be able to choose globally best algorithms based on objective, quantifiable methods. We introduce both deterministic and stochastic approaches: the former do not require any assumptions such as distributional information on the quality of the prediction, whereas the latter

help us capture the notion of risk, which is inherently tied to the stochasticity of the prediction oracle. Specifically, we consider the following measures:

**Distance measures** We evaluate the *distance* between the performance of the algorithm, and an *ideal* solution, i.e. an omniscient algorithm that knows the input, but is constrained by the same robustness requirement as the online algorithm. We focus on two distance metrics: i) The weighted *maximum* distance, which is defined as the weighted $L_\infty$-norm distance between the performance function of the algorithm and that of the ideal solution; here, the weight is a user-specified function that reflects how much, and what type of importance the designer assigns to prediction errors; and ii) The weighted *average* distance, which measures the aggregate distance between the algorithm and the ideal solution, averaged over the range of the prediction error.

Distance measures are inspired by tools such as Receiver Operating Characteristic (ROC) graphs (Fawcett, 2006), which describe the tradeoff between the true positive rates (TPR) and the false positive rates (FPR) of classifiers. Distance metrics between two ROC curves have been used as a comparison measure of classifiers. Moreover, weighted distances in ROC graphs can help emphasize critical regions: e.g., a user who is sensitive to false positives when FPR is low. This weighted approach has several applications in medical diagnostic systems (Li & Fine, 2010).

**Risk measures** Here, the motivation comes from the realization that Pareto-optimal and tolerance-based algorithms handle the risk of deviating from a perfect prediction in totally different ways. Namely, the former maximize the risk, while the latter seek to minimize it. This explains undesirable characteristics such as their brittleness and inefficiency, respectively. To formalize the notion of risk, we first introduce a stochastic prediction setting, where the oracle provides imperfect distributional information to the algorithm. We then introduce a novel analysis approach based on a risk measure that has been influential in decision theory, namely the *Conditional Value-at-Risk*, denoted by $\text{CVaR}_\alpha$. This value measures, informally, the expectation of a random loss/reward on its $(1 - \alpha)$-fraction of worst outcomes (Sarykalin et al., 2008). Here, $\alpha \in [0, 1)$ is a parameter that measures the *risk aversion* of the end user. We show how to obtain a parameterized analysis based on risk-aversion, which quantifies the trade-off between the performance of the algorithm and its risk.

Our techniques generalize previous approaches in learning-augmented algorithms. More precisely, in the context of distance measures, by choosing the weight to be equal to 1 only at the prediction point and zero otherwise, we recover the Pareto-optimal algorithms. For the risk-based analysis, we obtain a generalization of the *distributional* consistency-robustness tradeoffs of Diakonikolas et al. (2021), by introducing the notion of $\alpha$-*consistency*, where $\alpha$ is the risk parameter.

The paper is structured as follows. In Section 2, we formally present the decision-theoretic framework of our study, which we then apply to various problems. For ski rental (Section 3) we show how to find, among the infinitely many Pareto-optimal algorithms, the one that optimizes our metrics. For one-max search (Section 4) we show how to find, for any parameter $r$, an algorithm that likewise optimizes the metrics, among the infinitely many $r$-robust strategies. Last, for contract scheduling (Section 5), we show how to find, among the infinitely many schedules of optimal acceleration ratio, one that simultaneously optimizes each of our target metrics. In Section 6, we provide an experimental evaluation of our algorithms that demonstrates the attained performance improvements.

**Other related work** Elenter et al. (2024), addressed brittleness via a user-specified profile. This differs from our approach, in that our measures induce an explicit comparison to an ideal algorithm, and are thus true performance metrics, unlike Elenter et al. (2024) which does not allow for pairwise comparison of algorithms. The conditional value-at-risk was recently used in Christianson et al. (2024) in the design and analysis of *randomized* algorithms without predictions; however, no previous work has connected CVaR to the competitive analysis of learning-augmented algorithms.

## 2 DECISION-THEORETIC MODELS

In this section, we formalize our decision-theoretic framework. For definiteness, we assume cost-minimization problems (e.g., ski rental), however we note that the definitions can be extended straightforwardly to profit-maximization problems (e.g., one-max search and contract scheduling). We denote by $\text{OPT}(\sigma)$ the cost of an optimal offline algorithm on an input sequence $\sigma$.

## 2.1 DISTANCE-BASED ANALYSIS

We focus on problems with single-valued predictions. We denote by $x_\sigma$ some significant information on the input $\sigma$, and by $y \in \mathbb{R}$ its predicted value. For instance, in one-max search, $x_\sigma$ is the maximum price in $\sigma$. When $\sigma$ is implied from context, we will use $x$ for simplicity. The prediction *error* is defined as $\eta = |x_\sigma - y|$. The *range* of a prediction $y$, denoted by $R_y$, is defined as an interval $R_y = [\ell, u] \subseteq [0, \infty)$ such that $x_\sigma \in R_y$. This formulation allows us to study algorithms with a tolerance parameter. In particular, if $R_y = [(1-\delta)y, (1+\delta)y]$ where $\delta \in [0, 1]$, then we refer to algorithms that operate under this assumption as $\delta$-*tolerant* algorithms. We emphasize that this assumption of a bounded prediction error is not necessary in our framework, and unless specified, we consider the general case $R_y = [0, \infty)$. Namely, we use this assumption to be able to compare against known $\delta$-tolerant algorithms.

Given an online algorithm $A$, an input $\sigma$, and a prediction $y$, we denote by $A(\sigma, y)$ the *cost* incurred by $A$ on $\sigma$, using $y$. The *performance ratio* of $A$, denoted by $\mathrm{pr}(A, \sigma, y)$, is defined as the ratio $\frac{A(\sigma,y)}{\mathrm{OPT}(\sigma)}$. We define the *consistency* (resp. *robustness*) of $A$ as its worst-case performance ratio given an error-free (resp. adversarial) prediction. Formally, $\mathrm{cons}(A) = \sup_\sigma \mathrm{pr}(A, \sigma, x_\sigma)$ and $\mathrm{rob}(A) = \sup_{\sigma,y} \mathrm{pr}(A, \sigma, y)$. We say that $A$ is $r$-robust if it has robustness at most $r$.

To define our distance measures, we introduce the concept of an *ideal* solution. Given $r \geq 1$ and an input $\sigma$, we define by $\mathrm{I}_r(\sigma)$ the smallest cost that can be achieved on $\sigma$ by an online algorithm $A$ that is required to be $r$-competitive on all inputs. We also define $\mathrm{pr}(\mathrm{I}_r, \sigma)$ as $\frac{\mathrm{I}_r(\sigma)}{\mathrm{OPT}(\sigma)}$. The definition implies that $\mathrm{I}_r$ is the *best-possible* Pareto-optimal algorithm with prediction $x_\sigma$. Note that any $r$-robust online algorithm $A$ with prediction $y$ obeys $\mathrm{pr}(A, \sigma, y) \geq \mathrm{pr}(I_r, \sigma)$.

We can now define our distance measures starting with the *maximum* weighted distance. Here, the user specifies a *weight* function $w_y : R_y \to [0, 1]$, which quantifies the importance that the user assigns to prediction errors, and aims to guarantee smoothness. To reflect this, we require that $w_y$ is piecewise monotone. Namely, if $R_y = [\ell, u]$, then $w_y$ is non-decreasing in $[\ell, y]$ and non-increasing in $[y, u]$. The maximum distance of an algorithm $A$, given $r, y$ is defined as

$$d_{\max}(A) = \sup_{\sigma, x \in R_y} \{(\mathrm{pr}(A, \sigma, x) - \mathrm{pr}(I_r, \sigma)) w_y(x)\}. \tag{1}$$

Thus, the maximum distance measures the weighted maximum deviation from the ideal performance. We also define the *average* weighted distance, which measures the average deviation from the ideal performance, across the range of the prediction error. Formally:

$$d_{\mathrm{avg}}(A) = \sup_\sigma \frac{1}{|R_y|} \int_{R_y} (\mathrm{pr}(A, \sigma, z) - \mathrm{pr}(I_r, \sigma)) w_y(z) \, dz. \tag{2}$$

## 2.2 RISK-BASED ANALYSIS

Since risk is an inherently stochastic concept, we need to introduce stochasticity in the prediction model. To this end, we assume that the prediction is in the form of a distribution $\mu$, with support over an interval $[\ell, u] \subseteq \mathbb{R}$, and a pdf that is non-decreasing on $[\ell, y]$ and non-increasing on $[y, u]$. This model has two possible interpretations. First, one may think of $\mu$ as a *distributional* prediction, in the lines of stochastic prediction oracles (Diakonikolas et al., 2021). A second interpretation of $\mu$ is that of a *prior* on the predicted value, based on historical data. We will use $R_\mu$ to refer to the range of $\mu$, since it is motivated by considerations similar to the notion of range in the distance measures.

Our analysis will rely on the Conditional Value-at Risk (CVaR) measure from the theory of risk management (Rockafellar et al., 2000). Let $X$ be a random variable that corresponds to the loss (e.g., the cost in the case of a minimization problem), and a parameter $\alpha \in [0, 1)$ that describes the risk *aversion*. The Conditional Value-at-Risk $\mathrm{CVaR}_\alpha$ is defined as

$$\mathrm{CVaR}_\alpha(X) = \inf_t \left\{ t + \frac{1}{1-\alpha} \mathbb{E}[(X-t)^+] \right\}, \quad \text{where} \quad (X-t)^+ = \max\{X-t, 0\}. \tag{3}$$

In words, $\mathrm{CVaR}_\alpha(X)$ is the expectation of $X$ on the $\alpha$-tail of its distribution, that is, the worst $(1-\alpha)$ fraction of its outcomes. Let $\mathcal{F}$ denote the class of *input* distributions (i.e., distributions

over sequences $\sigma$) in which the predicted information has the same distribution as $\mu$. For example, in one-max search, $F$ is a distribution of input sequences such that the maximum price is distributed according to $\mu$. Given $\alpha \in [0, 1)$, we define the $\alpha$-*consistency* of an algorithm $A$ as

$$\alpha\text{-cons}(A) = \sup_{F \in \mathcal{F}} \frac{\text{CVaR}_{\alpha,F}(A(\sigma))}{\mathbb{E}_{\sigma \sim F}[\text{OPT}(\sigma)]}, \tag{4}$$

where the subscript $F$ in the notation of CVaR signifies that $\sigma$ is generated according to $F$. Our objective is then summarized as follows. Given a robustness requirement $r$, and a risk parameter $\alpha$, we would like to find an $r$-robust algorithm of minimum $\alpha$-consistency.

This measure is a risk-inclusive generalization of consistency, and interpolates between two extreme cases. The first case, when $\alpha = 0$, describes a *risk-seeking* algorithm that aims to minimize its expected loss without considering deviations from the distributional prediction. In this case, $\text{CVaR}_{\alpha,F}(A) = \mathbb{E}_{\sigma \sim F}[A(\sigma)]$, thus (4) is equivalent to the consistency of $A$ in the distributional prediction model of Diakonikolas et al. (2021). The second case, when $\alpha \to 1$, describes a *risk averse* algorithm: here, it follows that $\text{CVaR}_{\alpha,F}(A) = \sup_{\sigma \in \text{supp}(F)} A(\sigma)$, thus (4) describes the performance of $A$ in the adversarial situation in which all the probability mass is concentrated on a worst-case point within the prediction range. Note that this risk-based model is an adaptation of risk-sensitive randomized algorithms (Christianson et al., 2024) to learning-augmented settings.

## 3    SKI RENTAL

We consider the continuous version, in that skis can be bought at any time in $\mathbb{R}$. We denote by $b \geq 1$ the buying cost, and by $x$ the skiing horizon that is unknown to the online algorithm. We denote by $A_T$ the online algorithm that buys at time $T$, hence its cost, $A_T(x)$, is equal to $x$, if $x < T$, and $b + T$, if $x \geq T$. In the learning-augmented setting, the oracle provides a prediction $y$ on the horizon. It is known that for $r \geq 2$, $A_T$ is $r$-robust iff $T \in [b/(r-1), b(r-1)]$. Purohit et al. (2018b) and Wei & Zhang (2020) showed that $r$-robust Pareto-optimal algorithms have consistency $r/(r-1)$. More generally, Benomar & Perchet (2025), gave a *class* of Pareto-optimal algorithms, whose members exhibit different, and incomparable smoothness.

*Objective:* Given a robustness requirement $r$ and a prediction $y$ on the number of skiing days, find $T \in [b/(r-1), b(r-1)]$ such that $A_T$ minimizes the various objectives defined in Section 2. We will denote by $T^*_{\text{max}}$, $T^*_{\text{avg}}$ and $T^*_{\text{cvar}}$ the optimal thresholds according to the corresponding measures.

### 3.1    DISTANCE MEASURES

We begin by expressing the ideal performance.

**Lemma 1** (Appendix A). *The performance ratio of the ideal algorithm $I_r$ is*

$$\text{pr}(I_r, x) = \begin{cases} 1, & \text{if } x < b, \\ \frac{x}{b}, & \text{if } x \in \left[b, \min\left\{\frac{br}{r-1}, b(r-1)\right\}\right], \\ \frac{r}{r-1}, & \text{if } x \geq \min\left\{\frac{br}{r-1}, b(r-1)\right\}. \end{cases}$$

For some intuition behind the proof, we distinguish between three cases. If $x < b$, then $I_r$ buys at $b$. If $x > b$, it buys at $\min\left\{\frac{br}{r-1}, b(r-1)\right\}$; and if $x \geq \min\left\{\frac{br}{r-1}, b(r-1)\right\}$ it buys at $b/(r-1)$. These choices optimize its cost on input $x$, while guaranteeing $r$-robustness on all inputs. Note that $\text{pr}(I_r, x)$ has a discontinuity at $x = (r-1)b$ only if $\frac{r}{r-1} \geq r-1$, or $r < 2.618$, approximately. For simplicity, we will consider the case $r > 2.618$, for which the ideal performance is continuous, and we refer to the Appendix for a discussion of the case $r \in [2, 2.618]$.

Figure 1 illustrates the performance of the ideal algorithm (in black, bold line) and various online algorithms $A_T$. Note that all online algorithms have no better performance than the ideal on all inputs, as expected, and that no online algorithm dominates the others.

We will distinguish between online algorithms that buy at times in $[b/(r-1), b)$, and those that buy at times in $[b, b(r-1)]$; we denote these two classes by $C_{<b}$ and $C_{\geq b}$, respectively. This distinction

will be helpful in the computational optimization of the distance measures, namely in the proof of Theorem 2. From (1), given a prediction $y$ with range $R_y$ we have that

$$d_{\max}(A_T) = \sup_{x \in R_y} \left( \frac{A_T(x)}{\min\{x, b\}} - \mathrm{pr}(\mathrm{I}_r, x) \right) w(x). \tag{5}$$

To gain some insight into the structure of the maximum distance objective in (5), let us first consider the unweighted case, i.e., $w(x) = 1$. If $R_y$ is unbounded, then $d_{\max}^* = 1$, which is attained by all $A_T \in C_{\geq b}$. Note that among these algorithms, $A_b$ has the best consistency, so we may choose this algorithm as a tie-breaker.

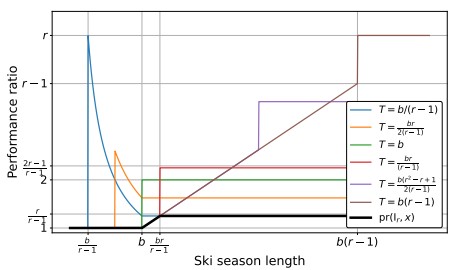

Figure 1: Performance functions of $r$-robust algorithms for different threshold values. The curve in bold depicts the ideal performance ratio.

That is, if we have no confidence on the quality of the prediction, the best algorithm is the competitively optimal one, which agrees with intuition. If, however, $R_y$ has known bounds, then the best algorithm depends on this range. For instance, if the right endpoint $u$ of $R_y$ is smaller than $b$, then any algorithm in the class $C_{\geq b}$ is optimal according to $d_{\max}$. This is consistent with all algorithms in the class defined by Benomar & Perchet (2025). It is also interesting to note that regardless of the range $R_y$, the best algorithm according to $d_{\max}$ is either a single algorithm in $C_{<b}$, namely $A_b$, or a choice of algorithms in $C_{\geq b}$; this is precisely the set of all algorithms in Benomar & Perchet (2025). The following theorem shows how to obtain the best threshold for the weighted distance.

**Theorem 2** (Appendix A). *There is an algorithm for computing $T_{\max}^*$ that runs in time linear in the number of critical points of the weight function $w$ in the range $R_y$.*

We now turn to the average distance objective, which from (2) is equal to

$$d_{\mathrm{avg}}(A_T) = \frac{1}{|R_y|} \int_{R_y} \left( \frac{A_T(z, y)}{\min\{z, b\}} - \mathrm{pr}(\mathrm{I}_r, z) \right) w(z) \, dz. \tag{6}$$

In the unweighted case ($w = 1$), and without assumptions on $R_y$, $A_{b(r-1)}$ minimizes the average distance. This is because, for $x \geq \frac{br}{r-1}$, its performance ratio matches the ideal, as depicted by the blue curve in Figure 1. When $R_y$ is bounded, the optimal algorithm depends on the range, and the problem reduces to minimizing the area between the performance curves of $A_T$ and $I_r$ over $R_y$. For general weight functions, the integral in (6) is evaluated piecewise, depending on $w$ and $T$.

## 3.2 RISK-BASED ANALYSIS

We consider the CVaR-based analysis of ski rental. In this setting, the algorithm has access to a distributional prediction $\mu$ over the skiing horizon $x$. Following the discussion in Section 2.2, we will evaluate an online algorithm $A_T$ by means of its $\alpha$-consistency (4).

Define $q_T = \Pr[A_T(x) = T + b] = \int_T^\infty \mu(x) dx$. With this definition in place, we obtain the following result.

**Theorem 3** (Appendix A). *Let $x \sim \mu$ and $t^*$ denote the $\alpha$-quantile of $\mu$, i.e., the value satisfying $\int_0^{t^*} \mu(z) \, dz = \alpha$. Then*

$$\mathrm{CVaR}_{\alpha,\mu}[A_T(x)] = \min \left\{ \frac{1}{1-\alpha} \left( \int_{t^*}^T z \, \mu(z) \, dz + (T + b) \, q_T \right), \ T + b \frac{q_T}{1-\alpha}, \ T + b \right\}, \tag{7}$$

*and $T_{\mathrm{CVaR}}^* = \arg\min_{T \in \left[ \frac{b}{r-1}, \ b(r-1) \right]} \mathrm{CVaR}_{\alpha,\mu}[A_T(x)]$.*

Theorem 3 captures the tradeoff between optimizing for average-case performance, on the one hand, and safeguarding against adversarial prediction, on the other hand. Specifically, if $\alpha = 0$, the

objective reduces to minimizing the expected cost, since $\int_0^T z \cdot \mu(z)\, dz + (b+T) \cdot q_T = \mathbb{E}_{z \sim \mu}[A_T(z)]$. In contrast, if $\alpha \to 1$, then we consider two cases: If $T$ is such that $q_T > 0$, then from (7), $\mathrm{CVaR}_{\alpha,\mu}[A_T(x)] = b + T$, whereas if $q_T = 0$, then $\mathrm{CVaR}_{\alpha,\mu}[A_T(x)] = u$. Hence, when $\alpha \to 1$, $\mathrm{CVaR}_{\alpha,\mu}[A_T(x)] = \min(b + T, u)$. The proof, and further details on these two extreme cases can be found in Appendix A.

## 4 ONE-MAX SEARCH

In this problem, the input is a sequence $\sigma$ of *prices* in $[1, M]$, where $M$ is known to the algorithm. We denote by $x_\sigma$ the maximum price in $\sigma$, or simply by $x$, when $\sigma$ is implied. Any online algorithm is a *threshold* algorithm, in that it selects some $T \in [1, M]$ and accepts the first price in $\sigma$ that is at least $T$. If such a price does not exist in $\sigma$, then the profit of the algorithm is defined to be equal to the smallest price, namely equal to 1. We denote by $A_T$ an online algorithm $A$ with threshold $T$, and by $A_T(\sigma)$ its profit on input $\sigma$. In the learning-augmented setting, the online algorithm has access to a *prediction* $y$, and the prediction error is defined as $\eta = |x_\sigma - y|$.

The optimal competitive ratio of the problem is equal to $\sqrt{M}$ El-Yaniv (1998). Moreover, for any $r \geq \sqrt{M}$, it is easy to show that $A_T$ is $r$-robust if and only $T \in [t_1, t_2]$, where $t_1 = M/r$ and $t_2 = r$. Thus, for any $r > \sqrt{M}$, there is an infinite number of $r$-robust algorithms.

*Objective.* Given a robustness requirement $r$, find the threshold $T$ that optimizes the measures of Section 2. We denote by $T_{\max}^*$, $T_{\mathrm{avg}}^*$ and $T_{\mathrm{CVaR}}^*$ the optimal threshold values.

### 4.1 DISTANCE MEASURES

We first describe the ideal solution.

**Lemma 4** (Appendix B). *Given a robustness requirement $r$, and a sequence $\sigma$, the ideal algorithm $I_r$ chooses the threshold $\min\{t_2, \max\{t_1, x_\sigma\}\}$. Its performance ratio is*

$$\mathrm{pr}(I_r, \sigma) = \begin{cases} x_\sigma, & \text{if } x_\sigma \in [1, t_1) \\ 1, & \text{if } x_\sigma \in [t_1, t_2] \\ \frac{x_\sigma}{t_2}, & \text{if } x_\sigma \in (t_2, M]. \end{cases} \tag{8}$$

From (1) and Lemma 4, it follows that $d_{\max}(A_T) = \sup_{\sigma, x \in R_y} \left( \frac{x}{A_T(\sigma)} - \mathrm{pr}(I_r, \sigma) \right) w(x)$.

We first give an analytical solution for unweighted maximum distance. We refer to Appendix B for the proof, and some intuition about the following result.

**Theorem 5.** *For uniform weights ($w = 1$), for all $x \in R_y = [\ell, u]$*

$$T_{\max}^* = \begin{cases} \min\{t_2, \max\{t_1, \sqrt{u}\}\}, & \text{if } u \leq t_2, \\ \min\{t_2, \max\{t_1, \tilde{T}\}\}, & \text{otherwise,} \end{cases}$$

*where $\tilde{T} = t_2 - u + \sqrt{(u - t_2)^2 + 4t_2^2 u}$.*

The case of general weight functions is much more complex, from a computational standpoint. In Appendix B.2 we obtain a formulation as a two-person *zero-sum game* between the algorithm (that chooses its threshold $T$) and the adversary (that chooses $x$). For instance, if $R_y \subseteq [t_1, t_2]$, then the payoff function of this game is $\max \left\{ \max_{x \geq T} \left( \frac{x}{T} - 1 \right) \cdot w(x), \max_{x < T} (T - 1) \cdot w(x) \right\}$. In general, it is not possible to obtain an analytical expression of the value of this game (over deterministic strategies) for all weight functions, but the game can be solved analytically for relatively simple functions. For instance, in Appendix B.2, we solve the game analytically assuming linear weight. The average weighted distance, on the other hand, can be optimized by piece-wise evaluation of an integral. We refer to the discussion in Appendix B.3, and an example based on linear weights.

### 4.2 RISK-BASED ANALYSIS

We consider the setting in which the algorithm has access to a distributional prediction $\mu$ with support in $[(1 - \delta)y, (1 + \delta)y]$, for some given $\delta$. This assumption is not required, but it allows

us to draw useful conclusions as we discuss at the end of the section. Given robustness $r$, and a risk value $\alpha \in [0, 1)$, we seek an $r$-robust algorithm that minimizes the $\alpha$-consistency. Since this is a profit-maximization problem, the definitions of CVaR and $\alpha$-consistency are slightly different than (3) and (4). Namely, we have $\text{CVaR}_\alpha(X) = \sup_t \left\{ t - \frac{1}{1-\alpha} \mathbb{E}[(t - X)^+] \right\}$ (Rockafellar et al., 2000) and $\alpha\text{-cons}(A) = \sup_{F \in \mathcal{F}} \left( \frac{\mathbb{E}_{\sigma \sim F}[\text{OPT}(\sigma)]}{\text{CVaR}_{\alpha, F}(A(\sigma))} \right)$.

We first show that the $\alpha$-consistency is determined by a worst-case distribution $F^*$. Here, $F^*$ consists of sequences of infinitesimally increasing prices from 1 up to $y$, followed by a last price equal to 1, where $y$ is drawn according to $\mu$.

**Lemma 6** (Appendix B). *For any algorithm $A_T$ it holds that $\alpha\text{-Cons}(A_T) = \frac{\mathbb{E}_{x \sim \mu}[x]}{CVaR_{\alpha, F^*}[A_T(\sigma)]}$.*

Since the numerator in the expression of Lemma 6 is independent of the algorithm, it suffices to find the threshold $T$ for which $\text{CVaR}_{\alpha, F^*}[A_T(\sigma)]$ is maximized. This is accomplished in the following theorem. Define $q_T = \Pr_{\sigma \sim F^*}[A_T(\sigma) = 1]$.

From the definition of $F^*$ and the fact that $T$ is the threshold of $A$, it follows that $q_T = \int_{(1-\delta)y}^T \mu(p) dp$.

**Theorem 7** (Appendix B). $T^*_{\text{CVaR}} = \arg\max_{T \in [t_1, t_2]} \left\{ \frac{T(1 - \alpha - q_T) + q_T}{1 - \alpha}, \; (1 - \delta)y \right\}$.

Theorem 7 interpolates between two cases. When $\alpha = 0$, the algorithm maximizes its expected profit, assuming that the maximum price in the input sequence has distribution $\mu$. In this case, we find an $r$-robust algorithm of optimal consistency, under the distributional setting of Diakonikolas et al. (2021) which is a novel contribution for the one-max search problem by itself. When $\alpha \to 1$, the online algorithm must choose its threshold under the assumption that the maximum price is chosen adversarially within the support $[\ell, u]$ of $\mu$, hence the threshold (and the algorithm's profit) is $\ell = (1 - \delta)y$. This recovers the analysis of the $\delta$-tolerant algorithm in Angelopoulos et al. (2022).

## 5    Contract Scheduling

We apply our framework to the contract scheduling problem, defined in Section 1. Once again, the starting motivation is that there is an infinite number of schedules, all of which achieve an optimal robustness equal to 4 (Russell & Zilberstein, 1991). We show how to compute schedules that remain 4-robust, and provably optimize each of our metrics. Specifically, for maximum distance, the schedule is computed by considering the set of critical points (conceptually similar to Theorem 2); for average distance we derive a closed-form expression and optimize it numerically; and for CVaR we obtain an exact formula based on the predicted distribution. We also evaluate our schedules experimentally and observe that they outperform the state-of-the-art Pareto-optimal and $\delta$-tolerant schedules of Angelopoulos & Kamali (2023b). We refer to Appendix C for the detailed discussion.

## 6    Experimental Evaluation

We evaluate our algorithms of Sections 3 and 4 which optimize the maximum and average distance as well as the CVaR. We refer to them as MAX, AVG and $\text{CVAR}_\alpha$.

**Baselines** For ski rental, we compare to the class of algorithms $\text{BP}_\rho$ of Benomar & Perchet (2025). $\text{BP}_\rho$ buys at time $b/(r-1)$, if $y \geq b$, and at time $\rho \in [b, b(r-1)]$, otherwise. In our experiments we consider three possible values for the parameter $\rho$, namely $\rho \in \{b, (r-1)b, b + \frac{br}{2}\}$. For one-max search we compare to two Pareto-optimal algorithms: The one of Sun et al. (2021a), denoted by $\text{PO}_1$, and the more straightforward algorithm, denoted by $\text{PO}_2$ (Angelopoulos & Kamali, 2023a) that sets its threshold to $T = \min\{t_2, \max\{t_1, y\}\}$, where $t_1 = M/r$ and $t_2 = r$. We also compare against the $\delta$-tolerant algorithm of Angelopoulos et al. (2022), denoted by $\delta$-Tol.

**Datasets** For ski rental, we set $b = 10$ and $r = 5$. The prediction $y$ is chosen such that $y \sim \text{Unif}[b/z, bz]$, where $z = 4$, and the prediction range $R_y$ is set to $[(1 - \delta)y, (1 + \delta)y]$ with $\delta = 0.9$. The horizon $x$ is generated u.a.r. in $R_y$. For one-max search, we set $M = 1000$ and $r = 100$, with $y \sim \text{Unif}[z, M/z]$ and $z = 10$, and the same definition of $R_y$. The input is the worst-case sequence

Table 1: Experimental results for ski rental.

| | MAX | AVG | CVAR$_\alpha$ | | | BP$_\rho$ | | |
| --- | --- | --- | --- | --- | --- | --- | --- | --- |
| | | | $\alpha = 0.1$ | $\alpha = 0.5$ | $\alpha = 0.9$ | $b$ | $b + br/2$ | $b(r-1)$ |
| **Avg perf. ratio** | 1.344 | 1.337 | 1.340 | 1.349 | 1.367 | 1.677 | 2.187 | 2.219 |
| **CI$_+$/CI$_-$** | +0.03/-0.03 | +0.03/-0.03 | +0.03/-0.03 | +0.03/-0.03 | +0.03/-0.04 | +0.05/-0.05 | +0.16/-0.17 | +0.17/-0.17 |
| **Exp. cost** | 11.241 | 11.187 | 11.173 | 11.215 | 11.316 | 16.987 | 21.234 | 20.958 |
| **CI$_+$/CI$_-$** | +0.48/-0.51 | +0.49/-0.51 | +0.50/-0.52 | +0.48/-0.54 | +0.48/-0.52 | +0.99/-1.09 | +2.01/-2.22 | +1.95/-2.00 |

Table 2: Experimental results for one-max search.

| | MAX | AVG | CVAR$_\alpha$ | | | $\delta$-TOL | PO$_1$ | PO$_2$ |
| --- | --- | --- | --- | --- | --- | --- | --- | --- |
| | | | $\alpha = 0.1$ | $\alpha = 0.5$ | $\alpha = 0.9$ | | | |
| **Avg perf. ratio** | 4.394 | 4.447 | 9.771 | 8.144 | 6.022 | 10.009 | 4.630 | 15.685 |
| **CI$_+$/CI$_-$** | +0.04/-0.05 | +0.06/-0.05 | +0.26/-0.26 | +0.20/-0.20 | +0.12/-0.12 | +0.00/-0.00 | +0.06/-0.06 | +0.45/-0.45 |
| **Exp. profit** | 15.614 | 20.528 | 35.795 | 34.402 | 27.500 | 5.475 | 13.904 | 27.986 |
| **CI$_+$/CI$_-$** | +0.30/-0.32 | +0.50/-0.48 | +1.06/-1.04 | +1.00/-1.08 | +0.81/-0.82 | +0.15/-0.17 | +0.16/-0.17 | +0.82/-0.80 |

of increasing prices up to the maximum price $x$, followed by a last price equal to 1, where $x$ is chosen u.a.r. in $R_y$. This is the standard class of inputs for evaluating worst-case performance (Sun et al., 2021a; Elenter et al., 2024). We refer to Appendix D for more experimental results on the parameters $r, \delta, z$, as well as for experiments on real data for one-max search.

**Evaluation** For both problems, we compute the average performance ratio, over all $x \in R_y$, and over 1000 repetitions on the choice of $y$. For MAX and AVG, we use the (uniquely defined) linear, symmetric function over $R_y$, whereas CVAR$_\alpha$ is evaluated under a Gaussian distribution $\mu$ truncated to $R_y$ and centered at $y$, with $\alpha \in \{0.1, 0.5, 0.9\}$. Furthermore, we compute the average expected cost/profit over $\mu$ for the two problems, respectively, where the averaging is over the choices of $y$. Tables 1 and 2 show the obtained average performance ratios and expected costs/profits. The tables also report the 95% confidence intervals (CIs).

**Discussion** For the ski rental problem, all our algorithms achieve better performance ratios and average costs than the baseline BP$_\rho$, for all choices of the parameter $\rho$. The performance ratios and average costs of CVAR$_\alpha$ increase with $\alpha$, as expected, since the higher the parameter $\alpha$, the more the algorithm hedges against unfavorable outcomes.

For one-max search, the baseline algorithms show marked performance differences. This is due to the choice of thresholds, with $\delta$-TOL and PO$_2$ tending to have the smallest and largest thresholds, respectively, whereas PO$_1$ chooses its threshold in between. Thus, PO$_2$ and $\delta$-TOL show highest/lowest brittleness to prediction errors, respectively, whereas PO$_1$ is more balanced. MAX and AVG exhibit better performance ratios than all baselines and better expected profits, with the exception of the highly brittle PO$_2$. In regards to the CVAR class, once again the expected profit is decreasing with $\alpha$, whereas the performance ratio improves as $\alpha$ grows. We observe that CVaR algorithms considerably improve upon both $\delta$-TOL and PO$_2$ across both metrics. They also have a better average profit than PO$_1$, though worse performance ratio.

The experiments show that even with relatively simple weight functions and distance measures, distance-based algorithms offer considerable improvements over the state of the art. Moreover, CVaR approaches help obtain smooth tradeoffs between the expected cost/profit and the performance ratio, as a function of the risk parameter $\alpha$, with improved overall performance in the majority of the cases. In Appendix D we report further experimental results that allow us to reach additional conclusions on the impact of the various parameters in the setting. Specifically, as the prediction range becomes smaller, or as the weight function becomes more concentrated around $y$, the experiments show that the performance of our algorithms improves. This is consistent with the theoretical motivation and analysis, since they can better leverage information on the quality of the prediction.

## 7 Conclusion

We introduced new decision-theoretic approaches for optimizing the performance of learning-augmented algorithms, by taking into consideration the entire range of the prediction error. Future work can address further applications, e.g., generalized rent-or-buy problems such as multi-shop (Wang et al., 2020b) and multi-option ski rental (Shin et al., 2023), knapsack (Daneshvaramoli et al., 2024) and secretary problems (Antoniadis et al., 2023). Another direction involves problems with multi-valued predictions, such as packing problems (Im et al., 2021). Our framework can still apply in these more complex settings, since the error is defined by a distance norm between the predicted and the actual vector. A last direction concerns dynamic predictions, in which the oracle is accessed several times during the algorithm's execution. An interesting potential application in this domain is learning-augmented power management, given its connections to ski rental as shown in (Antoniadis et al., 2021).

**Ethics Statement.** This work introduces and studies theoretical measures for the design and analysis of learning-augmented online algorithms. We do not anticipate any ethical concerns arising from this research.

**Reproducibility Statement.** Concerning the theoretical results, all complete proofs are given in the Appendix. Concerning the experimental results, the full code and datasets can be downloaded at the `https://anonymous.4open.science/r/decision_theoretic_code-09E7/`. The code is also available as supplementary material.

## Acknowledgements

This work was supported by the grant ANR-23-CE48-0010 PREDICTIONS from the French National Research Agency (ANR).

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

# Appendix

## A    DETAILS FROM SECTION 3

### A.1    OMITTED PROOFS

*Proof of Lemma 1.* First, recall that in order to be $r$-robust, any algorithm must buy skis no later than time $b(r-1)$ and no earlier than $b/(r-1)$. We consider the following possible cases.

**Case 1:** $x < b$.
In this case, $I_r$ rents up to time $b$. This guarantees $r$-robustness, and since $I_r(x) = x$, we have that $\mathtt{pr}(I_r, x) = 1$.

**Case 2:** $x \in \left[b, \ \min\left\{\frac{br}{r-1}, \ b(r-1)\right\}\right)$.
In this case, $I_r$ buys at time $\min\left\{\frac{br}{r-1}, \ b(r-1)\right\} > x$, so $I_r(x) = x$. This guarantees $r$-robustness, since the buy time lies in $[b/(r-1), \ b(r-1)]$.

Now consider any $r$-robust algorithm $A_T$, then we consider two possible cases on $T$.

- If $T < b$, then $A_T$ buys before $b$, and its cost is at least that of $A_{b/(r-1)}$, which is
$$\mathrm{cost}(A_{b/(r-1)}, x) = \frac{b}{r-1} + b = \frac{br}{r-1} > x.$$

- If $T \in [b, \ \min\left\{\frac{br}{r-1}, \ b(r-1)\right\}]$, then $\mathrm{cost}(A_T, x) = T + b \geq x$.

In both cases, $\mathrm{cost}(A_T, x) \geq I_r(x) = x$. Therefore, $\mathtt{pr}(I_r, x) = x/b$ is optimal among $r$-robust algorithms.

**Case 3:** $x \geq \min\left\{\frac{br}{r-1}, \ b(r-1)\right\}$.
In this case, $I_r$ buys at time $b/(r-1)$, which satisfies $r$-robustness and its cost is
$$I_r(x) = \frac{br}{r-1},$$
while the offline optimum is $b$.

Now consider any other $r$-robust algorithm $A_T$. If $T > b/(r-1)$, then $\mathrm{cost}(A_T, x) = T + b > \frac{br}{r-1} = I_r(x)$. If $T < b/(r-1)$, then $A_T$ is not $r$-robust.

Therefore, $I_r(x)$ has the minimum cost among all $r$-robust algorithms, and
$$\mathtt{pr}(I_r, x) = \frac{br}{(r-1)b} = \frac{r}{r-1}.$$

This completes the proof.    □

*Proof of Theorem 2.* For general weight functions $w(x)$, recall that we assume that $w$ is symmetric and piecewise monotone: i.e., non-decreasing for $x < y$ and non-increasing for $x > y$. The behavior of the maximum distance objective in (5) depends on the location of $y$ relative to $b$, and also on the buying threshold $T$. Recall from Lemma 1 that $\mathtt{pr}(I_r, x) = 1$ for $x < b$, and is increasing for $x \geq b$. Thus, algorithms in the class $C_{<b}$ incur strictly positive distance in $x < b$, while algorithms from $C_{\geq b}$ match the ideal in this interval, and may perform better when the weight is concentrated around $y < b$.

Given this structure, we partition the problem into two subproblems: computing the best threshold $T_1^*$ among algorithms in $C_{<b}$ and the best threshold $T_2^*$ in $C_{\geq b}$. Once the buy times $T_1^*$ and $T_2^*$ are computed, the final choice is
$$T_{\max}^* = \underset{T \in \{T_1^*, T_2^*\}}{\arg\min} \ d_{\max}(A_T).$$

---

**Algorithm 1** Algorithm for computing $T_{\max}^*$

---

**Input:** Prediction $y$ with range $R_y$, weight $w(x)$, robustness $r$, buy cost $b$.
**Output:** Optimal buy time $T_{\max}^*$ minimizing max distance.
    **Case 1:** $T \in C_{<b}$
1: Define critical set
$$S_1 \leftarrow \{y,\, b\} \cup \left\{ z \in R_y : \tfrac{d}{dz}\left( \left( \tfrac{A_T(z)}{z} - 1 \right) w(z) \right) = 0 \right\}$$

2: **for** $T \in S_1$ **do**
3:     Compute $d_{\max}(A_T) \leftarrow \max_{x \in R_y} \left( \tfrac{\text{cost}(A_T, x)}{x} - 1 \right) w(x)$
4: **end for**
5: $T_1^* \leftarrow \arg\min_{T \in S_1} d_{\max}(A_T)$
    **Case 2:** $T \in C_{\geq b}$
6: Define critical set
$$S_2 \leftarrow \{y,\, b,\, \tfrac{br}{r-1},\, b(r-1)\} \cup \left\{ z \in R_y : \tfrac{d}{dz}\left( \left( \tfrac{A_T(z)}{b} - \text{pr}(I_r, z) \right) w(z) \right) = 0 \right\}$$

7: **for** $T \in S_2$ **do**
8:     Compute $d_{\max}(A_T) \leftarrow \max_{x \in R_y} \left( \tfrac{\text{cost}(A_T, x)}{b} - \text{pr}(I_r, x) \right) w(x)$
9: **end for**
10: $T_2^* \leftarrow \arg\min_{T \in S_2} d_{\max}(A_T)$
11: **return** $T_{\max}^* \leftarrow \arg\min\{ d_{\max}(A_{T_1^*}),\ d_{\max}(A_{T_2^*}) \}$

---

Algorithm 1 shows how to compute $T_{\max}^*$. The algorithm evaluates a finite set of buy times based on critical points, which are values of $x$ where the weighted distance function may reach its maximum. These include the prediction $y$, the point $x = b$, the values $\frac{br}{r-1}$ and $b(r-1)$, and all solutions to the equation where the derivative of the weighted distance is zero.

$\square$

*Proof of Theorem 3.* We use the cost-minimization version of the Conditional Value-at-Risk, given by (3).

Fix $T \in \left[ \frac{b}{r-1}, b(r-1) \right]$ and let $x \sim \mu$ denote the predicted skiing horizon. Let $M(t) := \int_0^t \mu(z)\, dz$ be the cumulative distribution function, and $q_T = 1 - M(T)$ be the probability of the horizon being at least $T$. The cost of algorithm $A_T$ is

$$A_T(x) = \begin{cases} x, & x < T, \\ T + b, & x \geq T. \end{cases}$$

We evaluate $\text{CVaR}_{\alpha,\mu}[A_T(x)]$ by considering three possible ranges of $t$ in the definition above.

**Case 1:** $t < T$. In this case,

$$(A_T(x) - t)^+ = \begin{cases} x - t, & t \leq x < T, \\ T + b - t, & x \geq T, \\ 0, & x < t. \end{cases}$$

Multiplying the CVaR expression by $(1 - \alpha)$ gives

$$(1 - \alpha)\, \text{CVaR}_\alpha(A_T(x)) = \inf_{t > 0} \left\{ (1 - \alpha)t + \int_t^T (z - t)\, \mu(z)\, dz + (T + b - t) \int_T^\infty \mu(z)\, dz \right\}$$

$$= \inf_{t > 0} \left\{ (1 - \alpha)t + \int_t^T z\, \mu(z)\, dz + (T + b)\, q_T - t\,(1 - M(t)) \right\}. \quad (9)$$

Differentiating (9) with respect to $t$ gives

$$(1 - \alpha) - (1 - M(t)) = M(t) - \alpha.$$

Since the second derivative of (9) equals $\mu(t) \geq 0$, the minimizer is any $t^*$ satisfying $M(t^*) = \alpha$. If $M$ is continuous and strictly increasing, then $t^* = M^{-1}(\alpha)$ is unique. Substituting $t^*$ and using

$1 - M(t^*) = 1 - \alpha$ yields

$$\mathrm{CVaR}_\alpha(A_T(x)) = \frac{1}{1-\alpha}\left(\int_{t^*}^{T} z\,\mu(z)\,dz + (T+b)\,q_T\right). \tag{10}$$

**Case 2:** $T \le t \le T + b$. Here $(A_T(x) - t)^+ = 0$ for $x < T$ and $(T + b - t)$ for $x \ge T$. Since (3) is linear in $t$, the minimum occurs at one of the endpoints of the range of $t$. As a result, we have

$$\mathrm{CVaR}_\alpha(A_T(x)) = \min\left\{T + b\,\frac{q_T}{1-\alpha},\ T + b\right\}. \tag{11}$$

**Case 3:** $t \ge T + b$. In this range $(A_T(x) - t)^+ = 0$ for all $x$, so the infimum is attained at $t = T + b$, yielding

$$\mathrm{CVaR}_\alpha(A_T(x)) = T + b.$$

Combining the above cases, $\mathrm{CVaR}_{\alpha,\mu}[A_T(x)]$ is the minimum between (10) (Case 1) and (11) (Case 2), which completes the proof.

**Analysis of extreme cases.**

(i) For $\alpha = 0$, we have $t^* = \inf \mathrm{supp}(\mu) = \ell$. From (10),

$$\mathrm{CVaR}_0(A_T(x)) = \int_{\ell}^{T} z\,\mu(z)\,dz + (T+b)q_T = \mathbb{E}[\,A_T(x)\,],$$

since $A_T(x) = x$ on $\{x < T\}$ and $A_T(x) = T + b$ on $\{x \ge T\}$. Moreover, $\mathrm{CVaR}_0(A_T)$ is equal to (10), because

$$\int_{\ell}^{T} z\,\mu(z)\,dz + (T+b)q_T \le T(1 - q_T) + (T+b)q_T = T + bq_T.$$

(ii) Suppose that $\alpha \to 1$, then we distinguish between cases $q_T > 0$ and $q_T = 0$. If $q_T > 0$, both (10) and the first term of (11) contain $(1-\alpha)^{-1}$ times a positive quantity; the minimum is therefore the remaining term $T + b$, so

$$\lim_{\alpha\to1} \mathrm{CVaR}_\alpha(A_T(x)) = T + b.$$

If $q_T = 0$ (equivalently $T > u := \sup \mathrm{supp}(\mu)$), then (10) reduces to

$$\frac{1}{1-\alpha}\int_{t^*}^{T} z\,\mu(z)\,dz = \frac{1}{1-\alpha}\int_{t^*}^{u} z\,\mu(z)\,dz = \mathbb{E}[\,x \mid x \ge t^*\,],$$

since $\Pr[x \ge t^*] = 1 - \alpha$ and $\mu \equiv 0$ on $(u, T]$. As $\alpha \to 1$, $t^* \to u$ and bounded convergence yields $\mathbb{E}[x \mid x \ge t^*] \to u$. In the same regime, (11) becomes $\min\{T, T + b\} = T \ge u$, so the minimum is given by (10) and

$$\lim_{\alpha\to1} \mathrm{CVaR}_\alpha(A_T(x)) = u.$$

Combining the above we obtain that $\lim_{\alpha\to1} \mathrm{CVaR}_\alpha(A_T(x)) = \max\{\,u,\ T + b\,\}$.

$\square$

### A.2 IDEAL PERFORMANCE FOR $r < 2.618$

If $r \in [2, 2.618]$, then the ideal algorithm $\mathrm{I}_r(x)$ has a discontinuity at $x = b(r - 1)$, as shown in Figure 1. We note that all results presented in the main paper hold regardless of the value of $r$, i.e., Theorem 2 and Algorithm 1 remain valid.

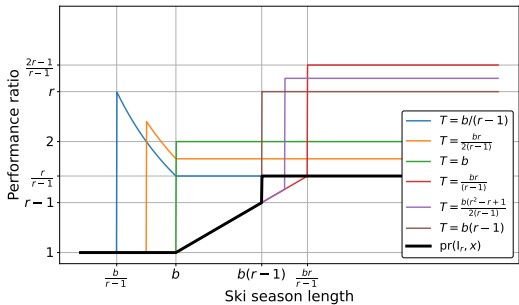

Figure 2: Performance of the ideal $r$-robust algorithm $I_r$ compared to several algorithms $A_T$ for different values of $T$, when $r < 2.618$. In contrast to Figure 1, the algorithms that buy at times $T = \frac{b(r^2 - r + 2)}{2(r-2)}$ (purple curve) and $T = \frac{br}{r-1}$ (red curve) are not $r$-robust anymore, because their thresholds fall outside the interval $[b/(r-1), b(r-1)]$.

## B    DETAILS FROM SECTION 4

### B.1    OMITTED PROOFS

*Proof of Lemma 4.* First, recall that in order to be $r$-robust, any algorithm must choose a threshold $T \in [t_1, t_2]$, where $t_1 = M/r$ and $t_2 = r$.

**Case 1:** $x_\sigma < t_1$.
In this case, $I_r$ sets $T = t_1$ and never accepts, so $A_{t_1}(\sigma) = 1$. The offline optimum is $x_\sigma$, hence

$$\text{pr}(I_r, \sigma) = \frac{x_\sigma}{1} = x_\sigma.$$

Any other $r$-robust algorithm $A_T$ must also have $T \geq t_1$, thus $A_T(\sigma) = 1$, and the performance ratio of $A_T$ is $x_\sigma$. Therefore, $I_r$ is optimal in this interval.

**Case 2:** $x_\sigma \in [t_1, t_2]$.
Here $I_r$ sets $T = x_\sigma$, thus $A_T(\sigma) = x_\sigma$ and

$$\text{pr}(I_r, \sigma) = \frac{x_\sigma}{x_\sigma} = 1,$$

hence $I_r$ is optimal in this interval.

**Case 3:** $x_\sigma > t_2$.
In this case, $I_r$ sets $T = t_2$ and accepts the first price at least $t_2$, so $A_{t_2}(\sigma) \geq t_2$ and

$$\text{pr}(I_r, \sigma) \leq \frac{x_\sigma}{t_2}.$$

Any $r$-robust algorithm must have $T \leq t_2$. If $T < t_2$, consider a sequence $\sigma$ of the form $v, t_2$, where $v \in [T, t_2)$. In this case, $A_T(\sigma) = v$ and

$$\text{pr}(A_T, \sigma) = \frac{x_\sigma}{v} \geq \frac{x_\sigma}{t_2}.$$

Thus no $r$-robust algorithm attains a smaller ratio than $I_r$ on $\sigma$.

Combining the three cases, the optimal threshold is $T = \min\{t_2, \max\{t_1, x_\sigma\}\}$ and the performance ratio is

$$\text{pr}(I_r, \sigma) = \begin{cases} x_\sigma, & x_\sigma \in [1, t_1), \\ 1, & x_\sigma \in [t_1, t_2], \\ x_\sigma/t_2, & x_\sigma \in (t_2, M]. \end{cases}$$

This completes the proof. $\qquad\square$

*Proof of Theorem 5.* The proof is based on a case analysis. First, note that the prediction range $R_y = [\ell, u]$ may not always overlap with the robustness interval $[t_1, t_2]$. If this is the case, the threshold $T^*_{\max}$ must be chosen so as to minimize the maximum distance from the ideal performance. Namely:

- If $u \leq t_1$, then $T^*_{\max} = t_1$, since in this case $d_{\max}(A_T) = 0$.

- If $\ell \geq t_2$, then $T^*_{\max} = t_2$, since we have again $d_{\max}(A_T) = 0$.

This ensures that the algorithm's performance aligns with the ideal benchmark when predictions fall outside the robustness interval. Furthermore, we analyze intersections between $R_y$ and $[t_1, t_2]$ by considering the following cases:

*Case 1: $\ell$ and $u$ are within $[t_1, t_2]$.* Then $\mathrm{pr}(I_r, x) = 1$ for all $x \in R_y$. We consider further subcases:

1. $T \leq \ell$: The maximum distance $d_{\max}(A_T)$ is defined by $\frac{u}{T} - 1$, with the adversary selecting $x = u$ to maximize this distance.

2. $T \in [\ell, u]$: The distance $d_{\max}(A_T)$ is calculated as $\max\left\{\frac{u}{T} - 1, T - 1\right\}$. If $T < x$, the performance ratio is maximized at $x = u$; however, when $T$ exceeds $x$, it is maximized at $x = T - \varepsilon$ for a very small $\varepsilon$. Hence in this case the performance ratio is arbitrarily close to $\frac{T}{1}$.

3. $T \geq u$: In this case, $x \leq u$, and $d_{\max}(A_T) = u - 1$.

The second case above, namely $T \in [\ell, u]$, is the most general one. To minimize $d_{\max}(A_T)$, the optimal $T^*_{\max}$ is equal to $\sqrt{u}$, because it minimizes the maximum of two expressions. If $\sqrt{u} < t_1$, then the threshold must be adjusted to:

$$T^*_{\max} = \max\{t_1, \sqrt{u}\}$$

to ensure it resides within the robustness interval.

*Case 2: $t_1 \leq \ell$ and $u \geq t_2$.* Here, the main complication is that $\mathrm{pr}(I_r, x)$ may differ from 1. It is sufficient to choose $T \in [\ell, t_2]$, with the maximum distance being

$$d_{\max}(A_T) = \max\left\{\frac{u}{T} - \frac{u}{t_2}, T - 1, \frac{t_2}{T} - 1\right\}.$$

Solving for the optimal $T$ that satisfies

$$\frac{u}{T} - \frac{u}{t_2} = T - 1,$$

yields

$$T = t_2 - u + \sqrt{(u - t_2)^2 + 4t_2^2 u}.$$

However, this value may not belong in $[t_1, t_2]$, hence

$$T^*_{\max} = \min\{t_2, \max\{t_1, T\}\}.$$

This concludes the proof. $\qquad\square$

For some intuition behind Theorem 5, we note that in the first case, the algorithm aims to minimize the distance from the line $y(x) = 1$ (the ideal performance). In this case, the threshold has a dependency on $\sqrt{u}$, as derived from an analysis similar to the competitive ratio (which is equal to the square root of the maximum price). In the second case, the algorithm aims to minimize the distance from a more complex ideal performance, which includes two line segments. This explains the dependency on the more complex value $\tilde{T}$. One can also show that $\tilde{T} \geq \sqrt{u}$: this is explained intuitively, since in the second case, the algorithm has more "leeway", given that the ideal performance ratio attains higher values.

*Proof of Lemma 6.* From the definition of $F^*$, it follows that

$$\mathbb{E}_{p \sim \mu}[p] = \mathbb{E}_{p \sim F^*}[p],$$

hence the $\alpha$-consistency of $A_T$ is at least the RHS of the equation in Lemma 6.

Let $\tilde{F}$ be a distribution that maximizes the $\alpha$-consistency, then it must be that

$$\alpha - \text{cons}(A_T) \geq \frac{\mathbb{E}_{p \sim \mu}[p]}{\text{CVaR}_{\alpha, \tilde{F}}[A_T(\sigma)]}.$$

We can argue that $\text{CVaR}_{\alpha, \tilde{F}}[A_T(\sigma)] \leq \text{CVaR}_{\alpha, F^*}[A_T(\sigma)]$. This follows directly from (3), and the observation that in any sequence $\sigma$ in the support of $F^*$, we have that $A_T(\sigma) = 1$, if $x_\sigma < T$, and $A_T(\sigma) = T$, if $x_\sigma \geq T$. Hence, we also showed that the $\alpha$-consistency is at least the RHS of the equation in Lemma 6, which concludes the proof. $\qquad\square$

*Proof of Theorem 7.* Similar to ski rental, the computation of $\text{CVaR}_{\alpha, \mu}[A_T(\sigma)]$ for the one-max search problem requires a case analysis based on the parameter $t$ of (3). Define $q_T = \Pr_{\sigma \sim F^*}[A_T(\sigma) = 1]$ as the probability that the algorithm $A_T$ selects the value 1, and $1 - q_T$ as the probability it selects the threshold $T$. Recall that these are the only two possibilities, from the definition of $F^*$, without any assumptions of $R_y$. Under the assumption that $R_y = [(1 - \delta)y, (1 + \delta)y]$, we can obtain a better lower bound for $A_T$, i.e., we know it can ensure a minimum profit of $(1 - \delta)y$. We proceed with the analysis of this setting, and consider the following cases.

*Case 1: $t \geq T$.* Then

$$\text{CVaR}_{\alpha, \mu}[A_T(\sigma)] = \sup_{t \geq T} \left\{ -t\left(\frac{\alpha}{1 - \alpha}\right) + \frac{q_T(1 - T) + T}{1 - \alpha} \right\}.$$

In this case, the optimal value of $t$ is equal to $T$, hence we obtain:

$$\text{CVaR}_{\alpha, \mu}[A_T(\sigma)] = \frac{T(1 - q_T - \alpha) + q_T}{1 - \alpha}.$$

*Case 2: $t \leq (1 - \delta)y$.* In this case, $(t - A_T(\sigma))^+ = 0$, and

$$\text{CVaR}_{\alpha, \mu}[A_T(\sigma)] = (1 - \delta)y.$$

*Case 3: $t \in [(1 - \delta)y, T]$.* Then

$$(t - A_T(\sigma))^+ = \begin{cases} 0, & \text{w. p. } 1 - q_T, \\ t - 1, & \text{w. p. } q_T, \end{cases}$$

from which we get that

$$\text{CVaR}_{\alpha, \mu}[A_T(\sigma)] = \sup_{t \in [(1 - \delta)y, T]} \left\{ t\left(\frac{1 - \alpha - q_T}{1 - \alpha}\right) + \frac{q_T}{1 - \alpha} \right\}.$$

We consider two further subcases. If $1 - \alpha - q_T \leq 0$, then we obtain

$$\text{CVaR}_{\alpha, \mu}[A_T(\sigma)] = (1 - \delta)y.$$

In the case, when $1 - \alpha - q_T > 0$, we have that

$$\text{CVaR}_{\alpha, \mu}[A_T(\sigma)] = \frac{T(1 - q_T - \alpha) + q_T}{1 - \alpha}.$$

Combining all the above cases, if follows that:

$$\text{CVaR}_{\alpha, \mu}[A_T(\sigma)] = \max \left\{ \frac{T(1 - q_T - \alpha) + q_T}{1 - \alpha}, (1 - \delta)y \right\}.$$

$\qquad\square$

## B.2 WEIGHTED MAXIMUM DISTANCE

The payoff function is defined considering two cases: First, if $R_y \subseteq [t_1, t_2]$, then the payoff function is

$$\max \left\{ \max_{x \geq T} \left( \frac{x}{T} - 1 \right) \cdot w(x), \max_{x < T} (T - 1) \cdot w(x) \right\},$$

since, in this case, the ideal performance is equal to 1.

In the second case, i.e., $R_y$ is not in $[t_1, t_2]$, then

$$\max \left\{ \max_{T \leq x \leq t_2} \left( \frac{x}{T} - 1 \right) w(x), \ \max_{x \geq t_2} \left( \frac{x}{T} - \frac{x}{t_2} \right) w(x), \ \max_{x < T} (T - 1) w(x) \right\}. \tag{12}$$

which follows from (8). In general, it is not possible to obtain an analytical expression for the value of this game (under deterministic strategies) for arbitrary weight functions. However, for specific functions, the game can be solvable as we demonstrate below.

**Example:** We will compute $T^*_{\max}$ for a linear weight function, defined as:

$$w(x) = \max \left\{ 0, 1 - \frac{|x - y|}{y\delta} \right\}. \tag{13}$$

For simplicity, we only show the computation for the case $R_y \subseteq [t_1, t_2]$. The other cases can be handled along similar lines, using (12). In this case (12) is

$$\max \begin{cases} (x - 1) \left( 1 - \dfrac{y - x}{y\delta} \right), & \text{if } x < T \text{ and } x \in [(1 - \delta)y, \ y], \\[2mm] \left( \dfrac{x}{T} - 1 \right) \left( 1 - \dfrac{y - x}{y\delta} \right), & \text{if } x \geq T \text{ and } x \in [(1 - \delta)y, \ y], \\[2mm] (x - 1) \left( 1 - \dfrac{x - y}{y\delta} \right), & \text{if } x < T \text{ and } x \in [y, \ (1 + \delta)y], \\[2mm] \left( \dfrac{x}{T} - 1 \right) \left( 1 - \dfrac{x - y}{y\delta} \right), & \text{if } x \geq T \text{ and } x \in [y, \ (1 + \delta)y]. \end{cases}$$

We denote the expressions, for each case in the above maximization, by $e_1, e_2, e_3, e_4$ respectively. First we analyze the best response of the adversary for a fixed threshold $T$, which represents the player's strategy. There are two cases to distinguish, depending on how $T$ compares to $y$.

CASE A: $T \leq y$.

**Subcase A1:** $(1 - \delta)y \leq x \leq T$   In this case, the value of the game is given by $e_1$. The second derivative of $e_1$ with respect to $x$ is $2/\delta y$, therefore $e_1$ is concave, and maximized at one of the endpoints of the case range. Considering $e_1$ as a function of $x$ we have $e_1((1 - \delta)y) = 0$ and $e_1(T) = (T - 1)(T - ((1 - \delta)y))/y\delta > 0$. Therefore, the adversary's best response is to choose $x = T$, yielding a game value, which we denote by

$$v_1 = (T - 1) \frac{T - ((1 - \delta)y)}{y\delta}.$$

**Subcase A2:** $T \leq x \leq y$   In this case, the value of the game is given by $e_2$. Its second derivative is $2/y\delta T$, therefore $e_2$ is concave. Again we evaluate $e_2$ at the endpoints of the case range, and obtain $e_2(T) = 0$ as well as $e_2(y) = y/T - 1 \geq 0$. Therefore, the adversary's best response is to choose $x = y$, producing a game value

$$v_2 = \frac{y}{T} - 1.$$

**Subcase A3:** $y \leq x \leq (1 + \delta)y$   In this case, the value of the game is given by $e_4$. The second derivative is $-2/(y\delta T)$, hence $e_2$ is concave. Using second order analysis, we find that it is maximized at $x = (T + (1 + \delta)y)/2$. This choice is in the case range $[y, (1 + \delta)y]$, since $T$ belongs to $[(1 - \delta)y, (1 + \delta)y]$. We denote by

$$v_4 = \frac{((1 + \delta)y - T)^2}{4y\delta T}$$

the value of the game for the best adversarial choice in this case.

**Summary of case A**   We observe that $v_2$ is always dominated by $v_4$, hence the value of the game in case $A$ is $\max\{v_1, v_4\}$.

CASE B: $T \geq y$.

Similar to the previous case, we break this case further into 3 subcases.

**Subcase B1:** $(1 - \delta)y \leq x \leq y$.   As in case $A1$, the value of the game is given by $e_1$, which is maximized at its right endpoint. Since this is a different endpoint than in case $A1$, we obtain a different value of the game, namely

$$e_1(y) = y - 1.$$

**Subcase B2:** $y \leq x \leq T$.   In this case, the value of the game is $e_3$. Its second derivative is $-2/y\delta$, hence it is concave. Its derivative at the upper endpoint is $4 - 2T \leq 0$, hence $e_2$ is maximized at this lower endpoint, and has the value $v_3 = y - 1$. Note this $v_3$ happens to be also the value of the game in case $B1$ and does not depend on $T$.

**Subcase B3:** $T \leq x$.   The analysis of this case is identical to the analysis of case $A4$, hence the value of the game is $v_4$.

**Summary of case B**   If the algorithm chooses $T \in [y, (1 + \delta)y]$, then the value of the game is $\max\{v_3, v_4\}$. We observe that $v_4$ is a concave function in $T$, with slope 0 at $T = (1 + \delta)y$, while $v_3$ is a constant. We show that $v_3 \geq v_4$, even for the whole range $(1 - \delta)y \leq T \leq (1 + \delta)y$. For this purpose we evaluate $v_4$ at $T = (1 - \delta)y$, and obtain by the assumption that $1 \leq (1 - \delta)y$ that

$$\begin{aligned}
v_3 - v_4((1 - \delta)y) &= y - 1 - \frac{y\delta}{(1 - \delta)y} \\
&\geq y - 1 - \frac{y\delta}{1} \\
&\geq 0.
\end{aligned}$$

SUMMARY OF BOTH CASES $A, B$

We know that if the algorithm chooses $T \geq y$, then the value of the game is $v_3 = y - 1$. We claim that $T \leq y$ would be a better choice. We already showed that $v_4 \leq v_3$. To show $v_1 \leq v_3$, we observe that in $v_1 = (T - 1)\frac{T - (y - y\delta)}{y\delta}$, the first factor $T - 1$ is upper bounded by $y - 1$. In addition, the second factor is at most 1 by $T \leq y$, from which we conclude $v_1 \leq v_3$.

Hence $\max\{v_1, v_4\} \leq v_3$. As a result, the algorithm's best strategy is to choose $(1 - \delta)y \leq T \leq y$ such that $v_1(T) = v_4(T)$. The exact expression of this value can be computed, but does not have a simple form. Hence, for the purpose of presentation, we omit its exact expression. See Figure 3 for an illustration.

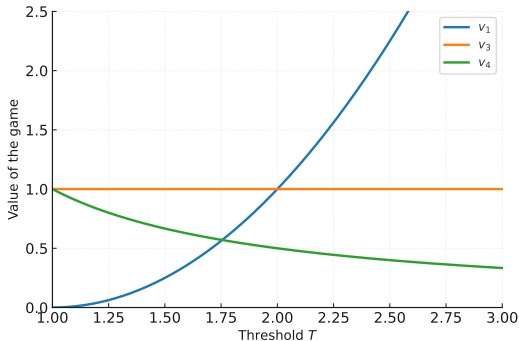

Figure 3: Illustration of the different values of the game: $v_1$ in blue, $v_3$ in orange, and $v_4$ in green. The parameters are $y = 2$ and $\delta = 0.5$. The value of the game is obtained for $T$ chosen as the intersection of the green and blue curves.

### B.3 An Example for Computing the Average Distance

In this section, we discuss how to optimize the average distance, which from (2), and (8) is equal to

$$
d_{\text{avg}}(A_T) = \begin{cases}
\frac{1}{2y\delta}(\int_{(1-\delta)y}^{T}(x-1) \cdot w(x) \, dx + \\
\int_{T}^{(1+\delta)y}(\frac{x}{T} - 1) \cdot w(x) \, dx), & \text{if } (1+\delta)y \leq t_2, \\
\frac{1}{2y\delta}(\int_{t_1}^{T}(x-1) \cdot w(x) \, dx + \\
\int_{T}^{t_2}(\frac{x}{T} - 1) \cdot w(x) \, dx + \\
\int_{t_2}^{(1+\delta)y}(\frac{x}{T} - \frac{x}{t_2}) \cdot w(x) \, dx), & \text{otherwise.}
\end{cases}
\tag{14}
$$

We illustrate how to compute the average distance for the linear weight function, as defined in (13). We show the calculations only for the first case in (14), i.e., in the case in which $(1 + \delta)y \leq t_2$. Recall that the linear weight function is increasing for $T \leq y$ and decreasing for $T \geq y$. Due to this behavior, we split the computation of the integral into two expressions, depending on whether $T < y$ or $T \geq y$, which are given below.

$$
\begin{aligned}
d_{\text{avg},1}(T) = \frac{1}{2y\delta} \Bigg( &\int_{(1-\delta)y}^{T}(x-1) \cdot \left(1 - \frac{y-x}{h}\right) dx \\
&+ \int_{T}^{y}\left(\frac{x}{T} - 1\right) \cdot \left(1 - \frac{y-x}{h}\right) dx \\
&+ \int_{y}^{(1+\delta)y}\left(\frac{x}{T} - 1\right) \cdot \left(1 - \frac{x-y}{h}\right) dx \Bigg),
\end{aligned}
$$

$$
\begin{aligned}
d_{\text{avg},2}(T) = \frac{1}{2y\delta} \Bigg( &\int_{(1-\delta)y}^{y}(x-1) \cdot \left(1 - \frac{y-x}{h}\right) dx \\
&+ \int_{y}^{T}\left(\frac{x}{T} - 1\right) \cdot \left(1 - \frac{x-y}{h}\right) dx \\
&+ \int_{T}^{(1+\delta)y}\left(\frac{x}{T} - 1\right) \cdot \left(1 - \frac{x-y}{y}\delta\right) dx \Bigg).
\end{aligned}
$$

The first expression, $d_{\text{avg},1}(T)$, can be simplified to:

$$d_{\text{avg},1}(T) = \frac{1}{12(y\delta)^2 T}\Big( -(y\delta)^3(T-1) + 3(y\delta)^2((y-2)T+y)$$
$$+ 3y\delta(T-1)\left(T^2 - y^2\right)$$
$$+ (T-1)(y-T)^2(y+2T)\Big).$$

Similarly, the second expression, $d_{\text{avg},2}(T)$, simplifies to:

$$d_{\text{avg},2}(T) = \frac{y\delta + 3y - (6 + y\delta - 3y)T}{12T}.$$

To determine the optimal threshold $T^*_{\text{avg}}$, we apply second-order analysis, solving for each case independently. The final solution is obtained by selecting the value of $T$ that minimizes $d_{\text{avg}}(T)$.

## C   APPLICATION IN CONTRACT SCHEDULING

**Definitions**   In this section, we apply our framework to the contract scheduling problem[1]. In its standard version (with no predictions), the schedule can be defined as an increasing sequence of the form $X = (x_i)_{i \in \mathbb{N}}$, where $x_i$ is the *length* of the $i$-th *contract*. These lengths correspond to the execution times of an interruptible system, i.e., we repeatedly execute the algorithm with running times $x_1, x_2, \dots$. Hence, the completion time of the $i$-th contract is defined as $\sum_{j=0}^{i} x_i$. Given an interruption time $T$, let $\ell(X, T)$ denote the length of the largest contract completed in $T$. The *acceleration ratio* of $X$ (Russell & Zilberstein, 1991) is defined as

$$acc(X) = \sup_T pr(X, T), \quad \text{where } pr(X, T) = \frac{T}{\ell(X, T)}. \tag{15}$$

It is known that the best-possible acceleration ratio is equal to 4, which is attained by any *doubling* schedule of the form $X_\lambda = (\lambda 2^i)_i$, where $\lambda \in [1, 2)$. In fact, under very mild assumptions, doubling schedules are the unique schedules that optimize the acceleration ratio. Note that according to Definition 15, without any assumptions, no schedule can have bounded acceleration ratio if the interruption time is allowed to be arbitrarily small. To circumvent this problem, it suffices to assume that the schedule is *bi-infinite*, in that it starts with an infinite number of infinitesimally small contracts. For instance, the doubling schedule can be described as $(2^i)_{i \in \mathbb{Z}}$, and the completion time of contract $i \geq 0$ is defined as $\sum_{j=-\infty}^{i} 2^j = 2^{i+1}$. We refer to the discussion in Angelopoulos et al. (2024) for further details. We summarize our objective as follows:

*Objective:* For each of the decision-theoretic models of Section 2, find $\lambda \in [1, 2)$ such that the schedule $X_\lambda$ optimizes the corresponding measure.

We will denote by $\lambda^*_{\max}$, $\lambda^*_{\text{avg}}$ and $\lambda^*_{\text{cvar}}$ the optimal values according to the maximum/average distance, and according to CVaR, respectively. Given a schedule $X_\lambda$, we will use the notation $k_\lambda(t)$ to denote the index of the largest contract in $X_\lambda$ that completes by time $t$, hence $k_\lambda(t) = \lfloor \log_2 \frac{t}{\lambda} \rfloor$.

### C.1   DISTANCE MEASURES

Here, we consider the setting in which there is a prediction $y$ on the interruption time. We begin with identifying an ideal schedule, which, in the context of contract schedule, is a 4-robust schedule $X$ that optimizes the length $\ell(X, y)$, i.e., the length of the contract completed by the predicted time $y$. From Angelopoulos & Kamali (2023b), we know that such an ideal schedule completes a contract of length $y/2$, precisely at time $y$, and thus has the following property.

**Remark 8.** *The performance ratio of the ideal 4-robust schedule is equal to 2.*

---

[1]This is a problem of incomplete information that can be considered as an online problem, in the sense that in each time step the scheduler must decide whether to continue the current contract, or start a new one.

---

**Algorithm 2** Algorithm for computing $\lambda^*_{\max}$

---

**Input:** Prediction $y$ with range $R_y$, weight function $w$.
**Output:** The optimal value of the $\lambda$-parameter, $\lambda^*_{\max}$.

1: Define a set of *critical* times as $S = \{y, \lambda \cdot 2^{k_\lambda(y)}, S'\}$, where $S'$ is the set of all solutions to the differential equation $w(T) + w'(T) \cdot (T - \lambda \cdot 2^{\lfloor \log_2 \frac{T}{\lambda} \rfloor}) = 0$.

2: For all $T \in S$, compute $d(X_\lambda, T) = \left( \frac{T}{\lambda \cdot 2^{k_\lambda(T)-1}} - 2 \right) \cdot w(T)$.

3: Return $\lambda^*_{\max} = \arg\min_{\lambda \in [1,2)} \max_{T \in S} d(X, T)$.

---

From (1), (15) and Remark 8 it follows that the maximum distance of a schedule $X_\lambda$ can be expressed as

$$d_{\max}(X_\lambda) = \sup_{T \in R_y} \left( \frac{T}{\ell(X_\lambda, T)} - 2 \right) \cdot w(T), \tag{16}$$

where recall that $R_y$ is the range of the prediction $y$.

Algorithm 2 shows how to compute $\lambda^*_{\max}$. We give the intuition behind the algorithm. We prove, in Theorem 9, that the distance can be maximized only at a discrete set of times, denoted by $S$. This set includes the prediction $y$, the last time a contract in $X_\lambda$ completes prior to $y$, and an additional set of times, denoted by $S'$ which are the roots of a differential equation, defined in step 2 of the algorithm. To show this, we rely on two facts: that $w$ is piece-wise monotone (i.e., bitonic), and that the performance function of any 4-robust schedule $X_\lambda$ is piece-wise linear, with values in $[2, 4]$.

**Theorem 9.** *Algorithm 2 returns an optimal schedule according to $d_{\max}$.*

*Proof.* Recall that the performance ratio of the schedule $X_\lambda = (\lambda 2^i)_i$ is expressed as

$$\mathrm{pr}(X_\lambda, T) = \frac{T}{\ell(X_\lambda, T)} = \frac{T}{\lambda \cdot 2^{\lfloor \log_2 \frac{T}{\lambda} \rfloor - 1}}.$$

We observe that $\mathrm{pr}(X_\lambda, T)$ is a piece-wise linear function. Specifically, if $T$ belongs in the interval $(\lambda 2^j, \lambda 2^{j+1}]$, then $\mathrm{pr}(X_\lambda, T)$ is a linear increasing function, with value equal to 2, at $T = \lambda 2^j + \varepsilon$, and value equal to 4 at $T = \lambda 2^{j+1}$, where $\varepsilon$ is an infinitesimally small, positive value. This linear growth arises from the structure of the schedule, which starts a new contract at the endpoint of each interval. For this reason, $\mathrm{pr}(X_\lambda, T)$ has a discontinuity at the endpoint of each interval.

By definition, $y$ belongs to the interval $(T_{k_\lambda(y)}, T_{k_\lambda(y)+1}]$. To simplify the notation, in the remainder of the proof we use $k$ to denote $k_\lambda(y)$. We claim $d_{\max}(X_\lambda)$ is maximized for some $T \in [T_k, T_{k+1}]$, specifically at one of a finite set of critical points $S$. To establish this claim, we make the following observations:

- At $T = T_k$, the performance ratio reaches its maximum value equal to 4, for the entire interval $(T_{k-1}, T_k]$.

- Any $T > T_{k+1}$ or $T < T_k$ does not need to be considered in the computation of $d_{\max}$, due to the monotonicity of the weight function, and the structural properties of the schedule $X_\lambda$, as discussed above.

Given that the bitonic nature of the weight function, we observe that for all $T < y$, $w(T)$ is non-decreasing, hence within the interval $[T_k, y)$, it suffices to only consider $T_k$ as a maximizing candidate. Furthermore, in the interval $[y, T_{k_\lambda+1}]$, $w(T)$ is non-increasing, while the performance ratio grows linearly. Thus, one must find the local maxima for $T \in (y, T_{k_\lambda+1}]$, by solving $d'(X_\lambda, T) = 0$, or equivalently

$$w(T) + w'(T) \cdot (T - \lambda \cdot 2^{\lfloor \log_2 \frac{T}{\lambda} \rfloor}) = 0.$$

We thus show that it suffices to consider the set $S$ as potential maximizers of the distance, as defined in Algorithm 2. $\square$

**Corollary 9.1.** *For the unit weight function $w(t) = 1$, and $R_y = [(1 - \delta)y, (1 + \delta)y]$, the schedule that minimizes $d_{\max}$ is the schedule of Angelopoulos & Kamali (2023b).*

*Proof.* The proof is a special case of the proof of Theorem 9. In this case, $d'(X_\lambda, T) = 1 > 0$, which implies that only local maxima for $d(X_\lambda, T)$ can occur at $T = T_{k+1}$ or at $(1 + \delta)y$.

We will consider two cases. First, suppose that $\delta > 1/3$. In this case, any schedule $X_\lambda$ is such that $d_{\max}(X_\lambda) = 2$. This is because $X_\lambda$ completes at least one contract within the time interval $[(1 - \delta)y, (1 + \delta)y]$.

For the second case, suppose that $\delta \leq 1/3$. Then, in order to minimize $d_{\max}$, and without loss of generality, $\lambda$ must be chosen so that no contract terminates anywhere in $[(1 - \delta)y, (1 + \delta)y]$, since otherwise $X_\lambda$ would have a performance ratio as large as 4, hence distance as large as 2. With this into account, $\lambda$ must be further chosen so that $X_\lambda$ completes a contract at time $(1 - \delta)y$. This is because, in this case, $\text{pr}(X_\lambda, T)$ is increasing in $T$, for $T \in [(1 - \delta)y, (1 + \delta)y]$. Hence the optimal algorithm is precisely the $\delta$-tolerant algorithm. $\qquad\square$

Next, we show how to optimize the average distance, which from (2), and (15) is equal to

$$d_{\text{avg}}(X) = \frac{1}{2y\delta} \int_{T \in R_y} \left( \frac{T}{\lambda \cdot 2^{\lfloor \log_2 \frac{T}{\lambda} \rfloor - 1}} - 2 \right) \cdot w(T)\, dT. \tag{17}$$

Optimizing (17) requires numerical methods.

## C.2    COMPUTING THE AVERAGE DISTANCE OF A SCHEDULE

To ensure computational tractability, we impose a constraint on the range $R_y$ of the prediction $y$. Specifically, we assume $\delta \leq \frac{1}{3}$. This assumption guarantees that for any schedule of the form $X = \lambda(2^i)_i$ there is at most one completed contract within $R_y$.

The length of the largest completed contract in $X$ before $(1 - \delta)y$ is then given by $\lambda 2^{k_\lambda((1-\delta)y)-1}$. Using this, we divide the range $R_y$ into two sub-intervals:

1. $[(1 - \delta)y, \lambda 2^{k_\lambda((1-\delta)y)+1}]$: In this interval, the performance ratio is

$$\frac{T}{\ell(X, T)} = \frac{T}{\lambda 2^{k_\lambda((1-\delta)y)-1}}.$$

2. $[\lambda 2^{k_\lambda((1-\delta)y)+1}, (1 + \delta)y]$: In this interval, the performance ratio is

$$\frac{T}{\ell(X, T)} = \frac{T}{\lambda 2^{k_\lambda((1-\delta)y)}}.$$

The average distance $d_{\text{avg}}(X)$ is then expressed as:

$$d_{\text{avg}}(X_\lambda) = \frac{1}{2y\delta} \left( \int_{(1-\delta)y}^{\lambda 2^{k_\lambda((1-\delta)y)+1}} \left( \frac{T}{\lambda 2^{k_\lambda((1-\delta)y)-1}} - 2 \right) \cdot w(T)\, dT \tag{18}$$

$$+ \int_{\lambda 2^{k_\lambda((1-\delta)y)+1}}^{(1+\delta)y} \left( \frac{T}{\lambda 2^{k_\lambda((1-\delta)y)}} - 2 \right) \cdot w(T)\, dT \right). \tag{19}$$

**Example: linear weight functions.** As an example, consider the case in which $w$ is a bitonic linear function defined by

$$w(T) = \max \left\{ 0, 1 - \frac{|T - y|}{y\delta} \right\},$$

To apply this weight function in the computation of (19), we divide the prediction interval $R_y = [(1 - \delta)y, (1 + \delta)y]$ into three subintervals based on the structure of the schedule and function $w$:

- $T \in [(1 - \delta)y, \lambda 2^{k_\lambda((1-\delta)y)+1}]$: In this case, $\text{pr}(X, T) = \frac{T}{\lambda 2^{k_\lambda((1-\delta)y)-1}}$. Then,

$$\int_{(1-\delta)y}^{\lambda 2^{k_\lambda((1-\delta)y)+1}} \left( \frac{T}{\lambda 2^{k_\lambda((1-\delta)y)-1}} - 2 \right) \cdot \left( 1 - \frac{y - T}{y\delta} \right) dT.$$

- $T \in [\lambda 2^{k_\lambda((1-\delta)y)+1}, y]$: In this case, $\mathrm{pr}(X,T) = \frac{T}{\lambda 2^{k_\lambda((1-\delta)y)}}$. Then,

$$\int_{\lambda 2^{k_\lambda((1-\delta)y)+1}}^{y} \left( \frac{T}{\lambda 2^{k_\lambda((1-\delta)y)}} - 2 \right) \cdot \left( 1 - \frac{y-T}{y\delta} \right) \, dT.$$

- $T \in [y, (1+\delta)y]$: In this case, $\mathrm{pr}(X,T) = \frac{T}{\lambda 2^{k_\lambda((1-\delta)y)}}$. Then,

$$\int_{y}^{(1+\delta)y} \left( \frac{T}{\lambda 2^{k_\lambda((1-\delta)y)}} - 2 \right) \cdot \left( 1 - \frac{y-T}{y\delta} \right) \, dT.$$

To summarize, we obtain from the above cases, and (19) that

$$d_{\mathrm{avg}}(X) = \frac{-3(y\delta)^2 \lambda + 4^{k_\lambda((1-\delta)y)+1}\lambda^3 + 3 \cdot 2^{k_\lambda((1-\delta)y)}\lambda^2 \, y(\delta - 1)}{3(y\delta)^2 \lambda}$$

$$+ \frac{2^{-2-k_\lambda((1-\delta)y)} \left( -(y\delta)^3 + 9(y\delta)^2 y - 3(y\delta)y^2 + y^3 \right)}{3(y\delta)^2 \lambda}.$$

Optimizing in terms of $\lambda$ via second-order analysis and solving for the derivative's root gives three solutions, but only one real root. Thus the optimized value is:

$$\lambda_{\mathrm{avg}}^* = 2^{-3(1+k_\lambda((1-\delta)y))} \left( 4^{k_\lambda((1-\delta)y)} \, y(1-\delta) \right.$$

$$+ \frac{16^{k_\lambda((1-\delta)y)} \, y^2(\delta-1)^2}{\left( -3 \cdot 64^{k_\lambda((1-\delta)y)}A + 4\sqrt{4096^{k_\lambda((1-\delta)y)}B_1 B_2} \right)^{1/3}}$$

$$+ \left. \left( -3 \cdot 64^{k_\lambda((1-\delta)y)}A + 4\sqrt{4096^{k_\lambda((1-\delta)y)}B_1 B_2} \right)^{1/3} \right).$$

where

$$A = 3(y\delta)^3 - 25(y\delta)^2 y + 9(y\delta)y^2 - 3y^3,$$
$$B_1 = 5(y\delta)^3 - 39(y\delta)^2 y + 15(y\delta)y^2 - 5y^3,$$
$$B_2 = (y\delta)^3 - 9(y\delta)^2 y + 3(y\delta)y^2 - y^3.$$

## C.3 RISK-BASED ANALYSIS

We now turn our attention to the CVaR analysis. Following the discussion of Section 2.2, the oracle provides the schedule with an imperfect distributional prediction $\mu$. From (4), and the fact that any distributional prediction concerns only the interruption time (the only unknown in the problem), the $\alpha$-consistency of a schedule $X_\lambda$ is equal to

$$\alpha\text{-cons}(A) = \frac{\mathbb{E}_{T \sim \mu}[T]}{\mathrm{CVaR}_{\alpha,\mu}[\ell(X_\lambda, T)]}.$$

We thus seek $X_\lambda$ that maximizes the conditional value-at-risk of its largest completed contract by an interruption generated according to $\mu$. To obtain a tractable expression of this quantity, we will assume that $\mu$ has support $R_y \in [(1-\delta)y, (1+\delta)y]$, where $h \leq y/3$. This captures the requirement that the support remains bounded, otherwise the distributional prediction becomes highly inaccurate. This implies that if $t$ is drawn from $\mu$, then in $X_\lambda$, $\ell(X_\lambda, t)$ can only have one of two possible values, namely $\lambda 2^{k_\lambda((1-\delta)y)-1}$ and $\lambda 2^{k_\lambda((1-\delta)y)}$.

Define $q_\lambda = \mathrm{Pr}[\ell(X_\lambda, T) = \lambda 2^{k_\lambda((1-\delta)y)-1}] = \int_{(1-\delta)y}^{\lambda 2^{k_\lambda((1-\delta)y)+1}} \mu(T)\,dT$, then from the discussion above we have that $\mathrm{Pr}[\ell(X_\lambda, T) = \lambda 2^{k_\lambda((1-\delta)y)}] = 1 - q_\lambda$. With this definition in place, we can find the optimal schedule.

**Theorem 10.** *Assuming $\delta \leq 1/3$, we have that*

$$\mathrm{CVaR}_{\alpha,\mu}[\ell(X_\lambda, T)] = \max\left\{\frac{\lambda 2^{k_\lambda((1-\delta)y)-1}}{1-\alpha}\left(2(1-\alpha)-q_\lambda\right), \lambda 2^{k_\lambda((1-\delta)y)-1}\right\},$$

*where $k_\lambda(t) = \lfloor \log_2 \frac{t}{\lambda} \rfloor$. Hence,*

$\lambda^*_{\mathrm{CVaR}} = \arg\max_{\lambda \in [1,2)} \mathrm{CVaR}_{\alpha,\mu}[\ell(X_\lambda, T)],$

Theorem 10 interpolates between two extreme cases. If $\alpha = 0$, then our schedule maximizes the expected contract length assuming $T \sim \mu$, i.e., $\lambda 2^{k_\lambda((1-\delta)y)-1} \cdot q_\lambda + \lambda 2^{k_\lambda((1-\delta)y)} \cdot (1-q_\lambda) = \lambda 2^{k_\lambda((1-\delta)y)-1} \cdot (2-q_\lambda)$. This schedule recovers the optimal consistency in the standard case of a distributional prediction, as studied in (Angelopoulos et al., 2024), and corresponds to a risk-seeking scheduler. In the other extreme, i.e., when $\alpha \to 1$, the schedule optimizes the length of a contract that completes by the time $(1-\delta)y$, namely $\lambda 2^{k_\lambda((1-\delta)y)-1}$. We thus recover the consistency of the $\delta$-tolerant schedule.

*Proof.* Recall the definition of the conditional value-at risk comes, as given in (3). In order to compute $\mathrm{CVaR}_{\alpha,\mu}[\ell(X_\lambda, T)]$ we have to apply case analysis, based on the value of the parameter $t$:

*Case 1: $t \geq \lambda 2^{k_\lambda((1-\delta)y)}$.* Then

$$\mathrm{CVaR}_{\alpha,\mu}[\ell(X_\lambda, T)] =$$
$$\sup_{t \geq \lambda 2^{k_\lambda((1-\delta)y)-1}}\left\{t - \frac{1}{1-\alpha}\left(t - \lambda 2^{k_\lambda((1-\delta)y)-1}(2-q_\lambda)\right)\right\}.$$

In this case, the optimal value of $t$ is equal to $\lambda 2^{k_\lambda((1-\delta)y)-1}$, hence we obtain:

$$\mathrm{CVaR}_{\alpha,\mu}[\ell(X_\lambda, T)] = \frac{\lambda 2^{k_\lambda((1-\delta)y)-1}}{1-\alpha}\left(2(1-\alpha)-q_\lambda\right).$$

*Case 2: $t \leq \lambda 2^{k_\lambda((1-\delta)y)-1}$.* In this case, $(t - \ell(X, T))^+ = 0$, and

$$\mathrm{CVaR}_{\alpha,\mu}[\ell(X_\lambda, T)] = \lambda 2^{k_\lambda((1-\delta)y)-1}.$$

*Case 3: $t \in [\lambda 2^{k_\lambda((1-\delta)y)-1}, \lambda 2^{k_\lambda((1-\delta)y)}]$.* Then

$$(t - \ell(X, T))^+ = \begin{cases} 0, & \text{w. p. } 1-q_\lambda, \\ t - \lambda 2^{k_\lambda((1-\delta)y)-1}, & \text{w. p. } q_\lambda, \end{cases}$$

from which we obtain that

$$\mathrm{CVaR}_{\alpha,\mu}[\ell(X_\lambda, T)] = \sup_{t \in [\lambda 2^{k_\lambda((1-\delta)y)-1}, \lambda 2^{k_\lambda((1-\delta)y)}]}\left\{t\left(1 - \frac{q_\lambda}{1-\alpha}\right) + \frac{\lambda 2^{k_\lambda((1-\delta)y)-1} \cdot q_\lambda}{1-\alpha}\right\}.$$

We consider two further subcases, based on whether of $1 - \alpha - q_\lambda$ is positive or not. In the former case, we have that

$$\mathrm{CVaR}_{\alpha,\mu}[\ell(X_\lambda, T)] = \frac{\lambda 2^{k_\lambda((1-\delta)y)-1}}{1-\alpha}\left(2(1-\alpha)-q_\lambda\right).$$

In the latter case, we obtain

$$\mathrm{CVaR}_{\alpha,\mu}[\ell(X_\lambda, T)] = \lambda 2^{k_\lambda((1-\delta)y)-1}.$$

From the above case analysis, it follows that

$$\mathrm{CVaR}_{\alpha,\mu}[\ell(X_\lambda, T)] = \max\left\{\frac{\lambda 2^{k_\lambda((1-\delta)y)-1}}{1-\alpha}\left(2(1-\alpha)-q_\lambda\right), \lambda 2^{k_\lambda((1-\delta)y)-1}\right\},$$

which concludes the proof. $\square$

Table 3: Experimental results for contract scheduling (linear weight), $\delta = 0.2$.

| | MAX | AVG | CVAR$_\alpha$ | | | PO | $\delta$-TOL |
|---|---|---|---|---|---|---|---|
| | | | $\alpha = 0.1$ | $\alpha = 0.5$ | $\alpha = 0.9$ | | |
| **Avg perf. ratio** | 2.421 | 2.413 | 2.426 | 2.416 | 2.471 | 2.960 | 2.500 |
| **CI$_+$/CI$_-$** | +0.0000/- 0.0000 | +0.0000/- 0.0000 | +0.0002/- 0.0002 | +0.0001/- 0.0001 | +0.0002/- 0.0003 | +0.0000/- 0.0000 | +0.0000/- 0.0000 |
| **Exp. contract length** | 413,303 | 417,504 | 420,164 | 415,274 | 402,646 | 373,008 | 397,875 |
| **CI$_+$/CI$_-$** | +9083.55/- 8957.96 | +8495.21/- 9051.60 | +9123.35/- 8027.55 | +8621.83/- 8983.43 | +8679.61/- 8684.86 | +7776.46/- 7024.64 | +8377.65/- 8357.20 |

Table 4: Experimental results for contract scheduling (linear weight), $\delta = \frac{1}{3}$.

| | MAX | AVG | CVAR$_\alpha$ | | | PO | $\delta$-TOL |
|---|---|---|---|---|---|---|---|
| | | | $\alpha = 0.1$ | $\alpha = 0.5$ | $\alpha = 0.9$ | | |
| **Avg perf. ratio** | 2.689 | 2.632 | 2.643 | 2.654 | 2.873 | 2.934 | 3.000 |
| **CI$_+$/CI$_-$** | +0.0000/- 0.0000 | +0.0000/- 0.0000 | +0.0002/- 0.0002 | +0.0002/- 0.0002 | +0.0006/- 0.0006 | +0.0000/- 0.0000 | +0.0000/- 0.0000 |
| **Exp. contract length** | 371,190 | 387,025 | 389,927 | 378,469 | 344,100 | 370,414 | 329,257 |
| **CI$_+$/CI$_-$** | +7982.66/- 7670.94 | +8086.65/- 8892.81 | +8739.76/- 9252.04 | +8779.84/- 7975.69 | +7710.75/- 8106.44 | +8252.32/- 8220.64 | +7185.13/- 7391.59 |

## C.4 EVALUATION OF CONTRACT SCHEDULES

**Baselines** We compare our schedules against two 4-robust baselines (Angelopoulos & Kamali, 2023b): the Pareto-optimal schedule (PO), which completes a contract at the prediction $y$, and the $\delta$-tolerant schedule ($\delta$-Tol), which completes a contract at $(1 - \delta)y$.

**Datasets** The prediction is chosen as $y \sim \mathrm{Unif}[0.8 \cdot 10^6, 1.2 \cdot 10^6]$, and the prediction range is $R_y = [(1 - \delta)y, (1 + \delta)y]$. The interruption time $T$ is chosen uniformly at random from $R_y$.

**Evaluation** For MAX and AVG, we use two weight functions defined on $R_y$. The first is a linear symmetric function that decreases from 1 at $y$ to 0 at the endpoints of $R_y$. The second is a Gaussian function

$$w(x) = \begin{cases} \frac{1}{\sigma\sqrt{2\pi}} \exp\left(-\frac{1}{2}\left(\frac{x-y}{\sigma}\right)^2\right), & \text{if } x \in R_y, \\ 0, & \text{otherwise,} \end{cases}$$

where $\sigma = \delta y/4$. For CVAR$_\alpha$, the predictive distribution $\mu$ coincides with the respective weight function, truncated and normalized on $R_y$, with $\alpha \in \{0.1, 0.5, 0.9\}$.

For each schedule ,we compute the average performance ratio, taken over all $T \in R_y$, with 1000 independent repetitions on the choice of $y$. We also compute the expected completed contract length under $\mu$, with the averaging performed over the choices of $y$. The tables also report the 95% confidence intervals (CIs). Results are presented in six tables: three for the linear weight function (Tables 3–5) and three for the Gaussian weight function (Tables 6–8), each corresponding to a different value of $\delta$.

**Discussion** The tables show that, in the vast majority of the considered settings, our schedules achieve better performance ratios and larger expected contract lengths than both PO and $\delta$-Tol. This can be explained by the fact that all Pareto-optimal algorithms are brittle, as shown in (Elenter et al., 2024), whereas, in contrast, the $\delta$-Tol algorithm is inefficient unless the prediction error is large.

For CVAR$_\alpha$, the results show that the expected completed contract length decreases as $\alpha$ grows, while the performance ratio tends to increase, which is consistent with the tradeoff between risk and robustness. When $\delta$ is smaller, all schedules perform better: this is because the reduced prediction range allows to our algorithms a more accurate positioning of the completion time. In a similar vein, for Gaussian weights, the results are consistently stronger than for linear weights, because the distribution is more concentrated around $y$. Finally, we note that most confidence intervals on the reported objectives collapse to zero. This is consistent with theory because the schedules have the same structure for each prediction $y$.

Table 5: Experimental results for contract scheduling (linear weight), $\delta = 0.4$.

|  | MAX | AVG | CVAR$_\alpha$ | | | PO | $\delta$-TOL |
|---|---|---|---|---|---|---|---|
|  |  |  | $\alpha = 0.1$ | $\alpha = 0.5$ | $\alpha = 0.9$ |  |  |
| **Avg perf. ratio** | 2.808 | 2.704 | 2.711 | 2.746 | 3.031 | 2.920 | 3.287 |
| **CI$_+$/CI$_-$** | +0.0000/-0.0000 | +0.0000/-0.0000 | +0.0001/-0.0001 | +0.0003/-0.0003 | +0.0003/-0.0003 | +0.0000/-0.0000 | +0.0000/-0.0000 |
| **Exp. contract length** | 361,661 | 385,593 | 387,891 | 372,668 | 335,125 | 376,514 | 308,199 |
| **CI$_+$/CI$_-$** | +7657.21/-7979.53 | +8265.97/-8367.98 | +8415.05/-8767.30 | +8449.57/-8393.17 | +7200.61/-7299.36 | +8415.63/-8840.35 | +7370.45/-8120.64 |

Table 6: Experimental results for contract scheduling (Gaussian weight), $\delta = 0.2$.

|  | MAX | AVG | CVAR$_\alpha$ | | | PO | $\delta$-TOL |
|---|---|---|---|---|---|---|---|
|  |  |  | $\alpha = 0.1$ | $\alpha = 0.5$ | $\alpha = 0.9$ |  |  |
| **Avg perf. ratio** | 2.267 | 2.266 | 2.273 | 2.266 | 2.315 | 2.960 | 2.500 |
| **CI$_+$/CI$_-$** | +0.0000/-0.0000 | +0.0001/-0.0001 | +0.0002/-0.0002 | +0.0000/-0.0001 | +0.0003/-0.0003 | +0.0000/-0.0000 | +0.0000/-0.0000 |
| **Exp. contract length** | 442,104 | 443,434 | 445,046 | 442,662 | 430,335 | 373,008 | 397,869 |
| **CI$_+$/CI$_-$** | +9722.63/-9586.90 | +9022.41/-9592.05 | +9673.17/-8505.70 | +9198.55/-9603.26 | +9289.41/-9245.24 | +7776.46/-7024.64 | +8377.52/-8357.07 |

Table 7: Experimental results for contract scheduling (Gaussian weight), $\delta = \frac{1}{3}$.

|  | MAX | AVG | CVAR$_\alpha$ | | | PO | $\delta$-TOL |
|---|---|---|---|---|---|---|---|
|  |  |  | $\alpha = 0.1$ | $\alpha = 0.5$ | $\alpha = 0.9$ |  |  |
| **Avg perf. ratio** | 2.421 | 2.413 | 2.423 | 2.415 | 2.524 | 2.934 | 3.000 |
| **CI$_+$/CI$_-$** | +0.0000/-0.0000 | +0.0001/-0.0001 | +0.0002/-0.0002 | +0.0001/-0.0001 | +0.0004/-0.0004 | +0.0000/-0.0000 | +0.0000/-0.0000 |
| **Exp. contract length** | 412,448 | 416,801 | 419,095 | 414,520 | 392,171 | 370,414 | 329,262 |
| **CI$_+$/CI$_-$** | +8872.45/-8524.03 | +8664.99/-9576.50 | +9395.10/-9951.62 | +9623.35/-8751.30 | +8729.12/-9156.69 | +8252.32/-8220.64 | +7185.24/-7391.71 |

Table 8: Experimental results for contract scheduling (Gaussian weight), $\delta = 0.4$.

|  | MAX | AVG | CVAR$_\alpha$ | | | PO | $\delta$-TOL |
|---|---|---|---|---|---|---|---|
|  |  |  | $\alpha = 0.1$ | $\alpha = 0.5$ | $\alpha = 0.9$ |  |  |
| **Avg perf. ratio** | 2.494 | 2.478 | 2.488 | 2.483 | 2.631 | 2.920 | 3.287 |
| **CI$_+$/CI$_-$** | +0.0000/-0.0000 | +0.0001/-0.0001 | +0.0003/-0.0002 | +0.0002/-0.0002 | +0.0006/-0.0006 | +0.0000/-0.0000 | +0.0000/-0.0000 |
| **Exp. contract length** | 407,466 | 413,970 | 416,511 | 410,594 | 382,442 | 376,514 | 308,032 |
| **CI$_+$/CI$_-$** | +8625.69/-8993.79 | +8887.56/-8941.28 | +9024.70/-9400.35 | +9319.17/-9231.67 | +8202.65/-8300.45 | +8415.63/-8840.35 | +6701.11/-7383.17 |

# D  DETAILS FROM SECTION 6

## D.1  EVALUATION OF SKI RENTAL ALGORITHMS

We base the experiments on the benchmarks described in Section 6, for varying values of the parameters $\delta, r, z$. For MAX and AVG we consider two classes of weight functions on $R_y = [(1 - \delta)y, (1 + \delta)y]$: a linear symmetric weight decreasing from 1 at $y$ to 0 at the endpoints, and a Gaussian weight with mean $y$ and $\sigma = \delta y/4$, both truncated and normalized on $R_y$. For CVAR$_\alpha$, the distribution $\mu$ is described by the same linear and Gaussian weight classes (truncated and normalized in $R_y$), with $\alpha \in \{0.1, 0.5, 0.9\}$. As in the main paper, we report (i) the average performance ratio (averaged over all $x \in R_y$ and over 1000 draws of $y$) and (ii) the expected cost under $\mu$ (averaged over the same 1000 draws of $y$). Each table also includes 95% confidence intervals. We present five tables for linear weights (Tables 9–13) and five for Gaussian weights (Tables 14–18). They correspond to the settings $(\delta, r, z) \in \{(0.9, 5, 4), (0.9, 8, 4), (0.5, 5, 4), (0.5, 8, 4), (0.9, 5, 7)\}$.

Table 9: Ski rental (linear weight), $\delta = 0.9$, $r = 5$, $z = 4$.

| | MAX | AVG | CVAR$_\alpha$ | | | BP$_\rho$ | | |
| --- | --- | --- | --- | --- | --- | --- | --- | --- |
| | | | $\alpha = 0.1$ | $\alpha = 0.5$ | $\alpha = 0.9$ | $b$ | $b + \frac{br}{2}$ | $b(r-1)$ |
| **Avg perf. ratio** | 1.3442 | 1.3368 | 1.3418 | 1.3637 | 1.3930 | 1.6767 | 2.1874 | 2.2189 |
| **CI$_+$/CI$_-$** | +0.0311/-0.0345 | +0.0309/-0.0336 | +0.0297/-0.0303 | +0.0343/-0.0342 | +0.0366/-0.0355 | +0.0482/-0.0502 | +0.1606/-0.1668 | +0.1706/-0.1684 |
| **Exp. cost** | 11.2501 | 11.1832 | 11.1693 | 11.2758 | 11.4553 | 16.4188 | 20.6070 | 20.5024 |
| **CI$_+$/CI$_-$** | +0.4798/-0.5076 | +0.4866/-0.5085 | +0.4954/-0.5201 | +0.4787/-0.5368 | +0.4671/-0.4981 | +0.9002/-0.9968 | +1.8565/-2.0281 | +1.8461/-1.8712 |

Table 10: Ski rental (linear weight), $\delta = 0.9$, $r = 8$, $z = 4$.

| | MAX | AVG | CVAR$_\alpha$ | | | BP$_\rho$ | | |
| --- | --- | --- | --- | --- | --- | --- | --- | --- |
| | | | $\alpha = 0.1$ | $\alpha = 0.5$ | $\alpha = 0.9$ | $b$ | $b + \frac{br}{2}$ | $b(r-1)$ |
| **Avg perf. ratio** | 1.2879 | 1.2853 | 1.3031 | 1.3343 | 1.3978 | 1.6767 | 2.2417 | 2.2251 |
| **CI$_+$/CI$_-$** | +0.0318/-0.0339 | +0.0324/-0.0334 | +0.0338/-0.0355 | +0.0435/-0.0398 | +0.0543/-0.0505 | +0.0482/-0.0502 | +0.1794/-0.1831 | +0.1748/-0.1732 |
| **Exp. cost** | 10.4306 | 10.3880 | 10.3999 | 10.4840 | 10.7296 | 16.4325 | 20.8787 | 20.7540 |
| **CI$_+$/CI$_-$** | +0.3959/-0.4364 | +0.4165/-0.4321 | +0.4189/-0.4595 | +0.4035/-0.4551 | +0.3811/-0.4256 | +0.9005/-0.9964 | +1.9688/-2.1559 | +1.9242/-1.9548 |

Table 11: Ski rental (linear weight), $\delta = 0.5$, $r = 5$, $z = 4$.

| | MAX | AVG | CVAR$_\alpha$ | | | BP$_\rho$ | | |
| --- | --- | --- | --- | --- | --- | --- | --- | --- |
| | | | $\alpha = 0.1$ | $\alpha = 0.5$ | $\alpha = 0.9$ | $b$ | $b + \frac{br}{2}$ | $b(r-1)$ |
| **Avg perf. ratio** | 1.2047 | 1.2034 | 1.2085 | 1.2145 | 1.2287 | 1.7729 | 2.2041 | 2.1976 |
| **CI$_+$/CI$_-$** | +0.0182/-0.0201 | +0.0182/-0.0204 | +0.0170/-0.0176 | +0.0182/-0.0198 | +0.0216/-0.0223 | +0.0633/-0.0695 | +0.1836/-0.1871 | +0.1870/-0.1844 |
| **Exp. cost** | 11.1739 | 11.1732 | 11.1751 | 11.2170 | 11.3174 | 17.0456 | 21.2944 | 21.0003 |
| **CI$_+$/CI$_-$** | +0.4695/-0.5043 | +0.4829/-0.5080 | +0.4960/-0.5209 | +0.4791/-0.5392 | +0.4821/-0.5183 | +0.9977/-1.1034 | +2.0426/-2.2322 | +1.9736/-2.0155 |

Table 12: Ski rental (linear weight), $\delta = 0.5$, $r = 8$, $z = 4$.

| | MAX | AVG | CVAR$_\alpha$ | | | BP$_\rho$ | | |
| --- | --- | --- | --- | --- | --- | --- | --- | --- |
| | | | $\alpha = 0.1$ | $\alpha = 0.5$ | $\alpha = 0.9$ | $b$ | $b + \frac{br}{2}$ | $b(r-1)$ |
| **Avg perf. ratio** | 1.1265 | 1.1245 | 1.1334 | 1.1404 | 1.1476 | 1.7729 | 2.1680 | 2.1595 |
| **CI$_+$/CI$_-$** | +0.0112/-0.0125 | +0.0108/-0.0124 | +0.0098/-0.0098 | +0.0133/-0.0128 | +0.0158/-0.0153 | +0.0633/-0.0695 | +0.1825/-0.1828 | +0.1783/-0.1757 |
| **Exp. cost** | 10.3897 | 10.3880 | 10.3983 | 10.4419 | 10.4889 | 17.0499 | 20.7854 | 20.7519 |
| **CI$_+$/CI$_-$** | +0.3918/-0.4351 | +0.4165/-0.4321 | +0.4184/-0.4550 | +0.3959/-0.4595 | +0.4063/-0.4330 | +0.9967/-1.1027 | +1.9364/-2.1260 | +1.9246/-1.9540 |

**Discussion**  As $\delta$ decreases, our algorithms perform better both in terms of the performance ratio and in terms of the average expected cost. This is consistent with theory, since $R_y$ becomes smaller, and the algorithms can better leverage the narrower prediction range. We also note that Gaussian weights generally yield stronger results than linear weights, which is explained by the fact that the Gaussian weight function is more concentrated around $y$.

Varying $z$ has small effect on the performance of the algorithms. This is not only expected, but also an essential feature of the algorithms, since they should perform consistently regardless of the predicted value. A similar observation holds for varying the robustness parameter $r$.

## D.2 EVALUATION OF ONE-MAX SEARCH ALGORITHMS

We base the experiments on the benchmarks described in Section 6, for varying values of the parameters $\delta, r, z$. We evaluate MAX, AVG, and CVAR$_\alpha$ using the linear and Gaussian weight classes

Table 13: Ski rental (linear weight), $\delta = 0.9$, $r = 5$, $z = 7$.

| | Max | Avg | CVAR$_\alpha$ | | | BP$_\rho$ | | |
| --- | --- | --- | --- | --- | --- | --- | --- | --- |
| | | | $\alpha = 0.1$ | $\alpha = 0.5$ | $\alpha = 0.9$ | $b$ | $b + \frac{br}{2}$ | $b(r-1)$ |
| **Avg perf. ratio** | 1.3103 | 1.3053 | 1.3151 | 1.3295 | 1.3534 | 1.7901 | 2.8509 | 2.9905 |
| **CI$_+$/CI$_-$** | +0.0229/-0.0222 | +0.0233/-0.0223 | +0.0238/-0.0260 | +0.0276/-0.0275 | +0.0312/-0.0310 | +0.0456/-0.0472 | +0.1914/-0.2102 | +0.2160/-0.2153 |
| **Exp. cost** | 11.7081 | 11.7023 | 11.7078 | 11.7775 | 11.9247 | 17.8170 | 27.2827 | 27.5432 |
| **CI$_+$/CI$_-$** | +0.3917/-0.4533 | +0.4131/-0.4432 | +0.4053/-0.4532 | +0.3772/-0.4741 | +0.3763/-0.4357 | +0.7726/-0.8356 | +2.0175/-2.3302 | +2.0996/-2.2661 |

Table 14: Ski rental (Gaussian weight), $\delta = 0.9$, $r = 5$, $z = 4$.

| | Max | Avg | CVAR$_\alpha$ | | | BP$_\rho$ | | |
| --- | --- | --- | --- | --- | --- | --- | --- | --- |
| | | | $\alpha = 0.1$ | $\alpha = 0.5$ | $\alpha = 0.9$ | $b$ | $b + \frac{br}{2}$ | $b(r-1)$ |
| **Avg perf. ratio** | 1.3514 | 1.3387 | 1.3397 | 1.3492 | 1.3669 | 1.6767 | 2.1874 | 2.2189 |
| **CI$_+$/CI$_-$** | +0.0297/-0.0333 | +0.0307/-0.0337 | +0.0302/-0.0311 | +0.0326/-0.0344 | +0.0335/-0.0358 | +0.0482/-0.0502 | +0.1606/-0.1668 | +0.1706/-0.1684 |
| **Exp. cost** | 11.1765 | 11.1731 | 11.1731 | 11.2151 | 11.3155 | 16.9873 | 21.2338 | 20.9577 |
| **CI$_+$/CI$_-$** | +0.4687/-0.5039 | +0.4829/-0.5081 | +0.4966/-0.5217 | +0.4796/-0.5398 | +0.4828/-0.5188 | +0.9850/-1.0909 | +2.0143/-2.2246 | +1.9520/-1.9971 |

Table 15: Ski rental (Gaussian weight), $\delta = 0.5$, $r = 5$, $z = 4$.

| | Max | Avg | CVAR$_\alpha$ | | | BP$_\rho$ | | |
| --- | --- | --- | --- | --- | --- | --- | --- | --- |
| | | | $\alpha = 0.1$ | $\alpha = 0.5$ | $\alpha = 0.9$ | $b$ | $b + \frac{br}{2}$ | $b(r-1)$ |
| **Avg perf. ratio** | 1.2097 | 1.2034 | 1.2049 | 1.2049 | 1.2141 | 1.7729 | 2.2041 | 2.1976 |
| **CI$_+$/CI$_-$** | +0.0184/-0.0206 | +0.0182/-0.0204 | +0.0180/-0.0185 | +0.0185/-0.0199 | +0.0197/-0.0221 | +0.0633/-0.0695 | +0.1836/-0.1871 | +0.1870/-0.1844 |
| **Exp. cost** | 11.1740 | 11.1732 | 11.1733 | 11.1733 | 11.2349 | 17.1177 | 21.3451 | 20.9444 |
| **CI$_+$/CI$_-$** | +0.4693/-0.5043 | +0.4829/-0.5080 | +0.4966/-0.5217 | +0.4843/-0.5294 | +0.4917/-0.5055 | +1.0257/-1.1153 | +2.0850/-2.2479 | +1.9824/-2.0130 |

Table 16: Ski rental (Gaussian weight), $\delta = 0.9$, $r = 8$, $z = 4$.

| | Max | Avg | CVAR$_\alpha$ | | | BP$_\rho$ | | |
| --- | --- | --- | --- | --- | --- | --- | --- | --- |
| | | | $\alpha = 0.1$ | $\alpha = 0.5$ | $\alpha = 0.9$ | $b$ | $b + \frac{br}{2}$ | $b(r-1)$ |
| **Avg perf. ratio** | 1.3025 | 1.2853 | 1.3002 | 1.3175 | 1.3571 | 1.6767 | 2.2417 | 2.2251 |
| **CI$_+$/CI$_-$** | +0.0322/-0.0335 | +0.0324/-0.0334 | +0.0342/-0.0360 | +0.0416/-0.0390 | +0.0507/-0.0444 | +0.0482/-0.0502 | +0.1794/-0.1831 | +0.1748/-0.1732 |
| **Exp. cost** | 10.3947 | 10.3880 | 10.3983 | 10.4381 | 10.5760 | 16.9918 | 20.7954 | 20.7520 |
| **CI$_+$/CI$_-$** | +0.3933/-0.4372 | +0.4165/-0.4321 | +0.4198/-0.4603 | +0.3974/-0.4614 | +0.4016/-0.4385 | +0.9841/-1.0901 | +1.9368/-2.1269 | +1.9246/-1.9540 |

Table 17: Ski rental (Gaussian weight), $\delta = 0.5$, $r = 8$, $z = 4$.

| | Max | Avg | CVAR$_\alpha$ | | | BP$_\rho$ | | |
| --- | --- | --- | --- | --- | --- | --- | --- | --- |
| | | | $\alpha = 0.1$ | $\alpha = 0.5$ | $\alpha = 0.9$ | $b$ | $b + \frac{br}{2}$ | $b(r-1)$ |
| **Avg perf. ratio** | 1.1345 | 1.1245 | 1.1288 | 1.1324 | 1.1404 | 1.7729 | 2.1680 | 2.1595 |
| **CI$_+$/CI$_-$** | +0.0147/-0.0149 | +0.0108/-0.0124 | +0.0103/-0.0105 | +0.0123/-0.0122 | +0.0152/-0.0150 | +0.0633/-0.0695 | +0.1825/-0.1828 | +0.1783/-0.1757 |
| **Exp. cost** | 10.3908 | 10.3880 | 10.3898 | 10.4073 | 10.4565 | 17.1216 | 20.7559 | 20.7519 |
| **CI$_+$/CI$_-$** | +0.3920/-0.4354 | +0.4165/-0.4321 | +0.4204/-0.4565 | +0.3986/-0.4622 | +0.4107/-0.4390 | +1.0255/-1.1160 | +1.9374/-2.1215 | +1.9246/-1.9540 |

defined in Section D.1. As in the main paper, we report (i) the average performance ratio (averaged over all $x \in R_y$ and over 1000 draws of $y$) and (ii) the expected profit under $\mu$ (averaged over the same 1000 draws). All tables include 95% confidence intervals. We present five tables for lin-

Table 18: Ski rental (Gaussian weight), $\delta = 0.9$, $r = 5$, $z = 7$.

| | MAX | AVG | CVAR$_\alpha$ | | | BP$_\rho$ | | |
| --- | --- | --- | --- | --- | --- | --- | --- | --- |
| | | | $\alpha = 0.1$ | $\alpha = 0.5$ | $\alpha = 0.9$ | $b$ | $b + \frac{br}{2}$ | $b(r-1)$ |
| **Avg perf. ratio** CI$_+$/CI$_-$ | 1.3201 +0.0244/- 0.0231 | 1.3065 +0.0236/- 0.0227 | 1.3110 +0.0238/- 0.0262 | 1.3199 +0.0262/- 0.0258 | 1.3383 +0.0313/- 0.0288 | 1.7901 +0.0456/- 0.0472 | 2.8509 +0.1914/- 0.2102 | 2.9905 +0.2160/- 0.2153 |
| **Exp. cost** CI$_+$/CI$_-$ | 11.7063 +0.3918/- 0.4544 | 11.7000 +0.4139/- 0.4431 | 11.7051 +0.4048/- 0.4595 | 11.7411 +0.3831/- 0.4670 | 11.8475 +0.4039/- 0.4368 | 18.2168 +0.7890/- 0.8573 | 30.3389 +2.4735/- 2.9018 | 30.9557 +2.6507/- 2.7672 |

Table 19: One-max (linear weight), $\delta = 0.9$, $r = 100$, $z = 10$.

| | MAX | AVG | CVAR$_\alpha$ | | | $\delta$-TOL | PO$_1$ | PO$_2$ |
| --- | --- | --- | --- | --- | --- | --- | --- | --- |
| | | | $\alpha = 0.1$ | $\alpha = 0.5$ | $\alpha = 0.9$ | | | |
| **Avg perf. ratio** CI$_+$/CI$_-$ | 4.3942 +0.04/-0.05 | 4.4469 +0.06/-0.05 | 9.9327 +0.26/-0.26 | 6.7335 +0.15/-0.15 | 5.2429 +0.12/-0.11 | 10.0085 +0.00/-0.00 | 4.6304 +0.06/-0.06 | 15.6848 +0.45/-0.45 |
| **Exp. profit** CI$_+$/CI$_-$ | 15.2276 +0.30/-0.33 | 19.6283 +0.49/-0.48 | 30.7586 +0.90/-0.89 | 27.7794 +0.80/-0.87 | 16.1624 +0.49/-0.50 | 5.4746 +0.15/-0.17 | 13.5565 +0.18/-0.18 | 27.9437 +0.82/-0.80 |

Table 20: One-max (linear weight), $\delta = 0.5$, $r = 100$, $z = 10$.

| | MAX | AVG | CVAR$_\alpha$ | | | $\delta$-TOL | PO$_1$ | PO$_2$ |
| --- | --- | --- | --- | --- | --- | --- | --- | --- |
| | | | $\alpha = 0.1$ | $\alpha = 0.5$ | $\alpha = 0.9$ | | | |
| **Avg perf. ratio** CI$_+$/CI$_-$ | 2.5877 +0.02/-0.02 | 2.7856 +0.01/-0.01 | 10.0269 +0.26/-0.26 | 6.7880 +0.16/-0.16 | 3.1589 +0.04/-0.04 | 2.0000 +0.00/-0.00 | 3.8718 +0.06/-0.07 | 21.0408 +0.62/-0.62 |
| **Exp. profit** CI$_+$/CI$_-$ | 28.6754 +0.79/-0.88 | 29.2159 +0.86/-0.82 | 36.1414 +1.07/-1.05 | 34.5976 +1.01/-1.09 | 29.7982 +0.86/-0.87 | 27.3730 +0.75/-0.83 | 13.9343 +0.16/-0.16 | 28.0349 +0.82/-0.81 |

Table 21: One-max (linear weight), $\delta = 0.9$, $r = 80$, $z = 10$.

| | MAX | AVG | CVAR$_\alpha$ | | | $\delta$-TOL | PO$_1$ | PO$_2$ |
| --- | --- | --- | --- | --- | --- | --- | --- | --- |
| | | | $\alpha = 0.1$ | $\alpha = 0.5$ | $\alpha = 0.9$ | | | |
| **Avg perf. ratio** CI$_+$/CI$_-$ | 4.4865 +0.04/-0.04 | 4.4469 +0.06/-0.05 | 9.9943 +0.26/-0.26 | 6.9624 +0.15/-0.15 | 5.7321 +0.14/-0.14 | 10.0085 +0.00/-0.00 | 4.6516 +0.05/-0.05 | 15.6848 +0.45/-0.45 |
| **Exp. profit** CI$_+$/CI$_-$ | 15.6165 +0.28/-0.31 | 19.6283 +0.49/-0.48 | 30.7121 +0.91/-0.89 | 27.6555 +0.81/-0.88 | 15.8462 +0.52/-0.53 | 5.4746 +0.15/-0.17 | 14.4844 +0.17/-0.18 | 27.9437 +0.82/-0.80 |

Table 22: One-max (linear weight), $\delta = 0.5$, $r = 80$, $z = 10$.

| | MAX | AVG | CVAR$_\alpha$ | | | $\delta$-TOL | PO$_1$ | PO$_2$ |
| --- | --- | --- | --- | --- | --- | --- | --- | --- |
| | | | $\alpha = 0.1$ | $\alpha = 0.5$ | $\alpha = 0.9$ | | | |
| **Avg perf. ratio** CI$_+$/CI$_-$ | 2.7301 +0.05/-0.05 | 2.7856 +0.01/-0.01 | 10.0378 +0.26/-0.26 | 6.7586 +0.16/-0.16 | 3.1362 +0.04/-0.04 | 2.0000 +0.00/-0.00 | 3.8570 +0.06/-0.07 | 21.0408 +0.62/-0.62 |
| **Exp. profit** CI$_+$/CI$_-$ | 28.6082 +0.78/-0.87 | 29.2159 +0.86/-0.82 | 36.0559 +1.07/-1.04 | 34.5122 +1.02/-1.10 | 29.7079 +0.86/-0.87 | 27.3730 +0.75/-0.83 | 14.9631 +0.16/-0.17 | 28.0349 +0.82/-0.81 |

ear weights (Tables 19–23) and five for Gaussian weights (Tables 24–28). They correspond to the settings $(\delta, r, z) \in \{(0.9, 100, 10), (0.5, 100, 10), (0.9, 80, 10), (0.5, 80, 10), (0.9, 100, 20)\}$.

**Discussion** The results show that our algorithms tend to improve as $\delta$ decreases, whereas they are not affected by variations in the parameters $r$ and $z$. This is consistent with theory, and we refer to the discussion in the analysis of the experiments on ski rental (Section D.1) for the justification.

For small values of $\delta$, $\delta$-TOL has very small performance ratio: this is due to the fact that in this case, the range is extremely small. This advantage disappears, in a marked manner, once $\delta$ increases.

Table 23: One-max (linear weight), $\delta = 0.9$, $r = 100$, $z = 20$.

| | MAX | AVG | CVAR$_\alpha$ | | | $\delta$-TOL | PO$_1$ | PO$_2$ |
| --- | --- | --- | --- | --- | --- | --- | --- | --- |
| | | | $\alpha = 0.1$ | $\alpha = 0.5$ | $\alpha = 0.9$ | | | |
| Avg perf. ratio | 3.8763 | 3.8645 | 6.7675 | 4.8622 | 4.1063 | 10.0049 | 3.8668 | 10.2558 |
| CI$_+$/CI$_-$ | +0.02/-0.02 | +0.02/-0.02 | +0.09/-0.09 | +0.05/-0.05 | +0.07/-0.07 | +0.00/-0.00 | +0.02/-0.02 | +0.15/-0.15 |
| Exp. profit | 11.2099 | 13.9404 | 19.7256 | 17.8093 | 10.9724 | 3.4915 | 11.7401 | 18.0238 |
| CI$_+$/CI$_-$ | +0.10/-0.10 | +0.18/-0.17 | +0.30/-0.30 | +0.27/-0.29 | +0.15/-0.15 | +0.05/-0.06 | +0.06/-0.06 | +0.28/-0.27 |

Table 24: One-max (Gaussian weight), $\delta = 0.9$, $r = 100$, $z = 10$.

| | MAX | AVG | CVAR$_\alpha$ | | | $\delta$-TOL | PO$_1$ | PO$_2$ |
| --- | --- | --- | --- | --- | --- | --- | --- | --- |
| | | | $\alpha = 0.1$ | $\alpha = 0.5$ | $\alpha = 0.9$ | | | |
| Avg perf. ratio | 5.0599 | 5.4540 | 9.7709 | 8.1444 | 6.0222 | 10.0085 | 4.6304 | 15.6848 |
| CI$_+$/CI$_-$ | +0.0728/-0.0756 | +0.0940/-0.0909 | +0.2564/-0.2561 | +0.1988/-0.1969 | +0.1196/-0.1205 | +0.0008/-0.0008 | +0.0561/-0.0578 | +0.4526/-0.4530 |
| Exp. profit | 24.9584 | 27.0416 | 35.7945 | 34.4015 | 27.5003 | 5.4746 | 13.9039 | 27.9858 |
| CI$_+$/CI$_-$ | +0.6259/-0.7080 | +0.7406/-0.7242 | +1.0580/-1.0403 | +1.0003/-1.0822 | +0.8137/-0.8159 | +0.1503/-0.1666 | +0.1583/-0.1659 | +0.8233/-0.8022 |

Table 25: One-max (Gaussian weight), $\delta = 0.5$, $r = 100$, $z = 10$.

| | MAX | AVG | CVAR$_\alpha$ | | | $\delta$-TOL | PO$_1$ | PO$_2$ |
| --- | --- | --- | --- | --- | --- | --- | --- | --- |
| | | | $\alpha = 0.1$ | $\alpha = 0.5$ | $\alpha = 0.9$ | | | |
| Avg perf. ratio | 5.9298 | 6.1761 | 12.1837 | 10.5279 | 7.4024 | 2.0000 | 3.8718 | 21.0408 |
| CI$_+$/CI$_-$ | +0.1000/-0.1051 | +0.1142/-0.1099 | +0.3366/-0.3318 | +0.2822/-0.2819 | +0.1767/-0.1844 | +0.0000/-0.0000 | +0.0622/-0.0653 | +0.6202/-0.6154 |
| Exp. profit | 35.2058 | 35.6679 | 41.5955 | 40.8581 | 37.3962 | 27.3730 | 13.9997 | 28.1314 |
| CI$_+$/CI$_-$ | +0.9612/-1.0665 | +1.0397/-1.0025 | +1.2372/-1.2094 | +1.1976/-1.2908 | +1.0807/-1.0969 | +0.7512/-0.8332 | +0.1485/-0.1558 | +0.8152/-0.8127 |

Table 26: One-max (Gaussian weight), $\delta = 0.9$, $r = 80$, $z = 10$.

| | MAX | AVG | CVAR$_\alpha$ | | | $\delta$-TOL | PO$_1$ | PO$_2$ |
| --- | --- | --- | --- | --- | --- | --- | --- | --- |
| | | | $\alpha = 0.1$ | $\alpha = 0.5$ | $\alpha = 0.9$ | | | |
| Avg perf. ratio | 5.1262 | 5.4540 | 9.8288 | 8.3568 | 6.2563 | 10.0085 | 4.6516 | 15.6848 |
| CI$_+$/CI$_-$ | +0.0622/-0.0690 | +0.0940/-0.0909 | +0.2528/-0.2506 | +0.1908/-0.1933 | +0.1229/-0.1208 | +0.0008/-0.0008 | +0.0465/-0.0473 | +0.4526/-0.4530 |
| Exp. profit | 24.8628 | 27.0416 | 35.6981 | 34.2272 | 27.3657 | 5.4746 | 14.9229 | 27.9858 |
| CI$_+$/CI$_-$ | +0.6067/-0.6877 | +0.7406/-0.7242 | +1.0671/-1.0390 | +1.0189/-1.0926 | +0.8320/-0.8334 | +0.1503/-0.1666 | +0.1671/-0.1722 | +0.8233/-0.8022 |

Table 27: One-max (Gaussian weight), $\delta = 0.5$, $r = 80$, $z = 10$.

| | MAX | AVG | CVAR$_\alpha$ | | | $\delta$-TOL | PO$_1$ | PO$_2$ |
| --- | --- | --- | --- | --- | --- | --- | --- | --- |
| | | | $\alpha = 0.1$ | $\alpha = 0.5$ | $\alpha = 0.9$ | | | |
| Avg perf. ratio | 5.9483 | 6.1761 | 12.1798 | 10.5404 | 7.3715 | 2.0000 | 3.8570 | 21.0408 |
| CI$_+$/CI$_-$ | +0.0807/-0.0907 | +0.1142/-0.1099 | +0.3329/-0.3376 | +0.2871/-0.2838 | +0.1781/-0.1906 | +0.0000/-0.0000 | +0.0629/-0.0660 | +0.6202/-0.6154 |
| Exp. profit | 34.9084 | 35.6679 | 41.4923 | 40.7649 | 37.2693 | 27.3730 | 15.0337 | 28.1314 |
| CI$_+$/CI$_-$ | +0.9659/-1.0843 | +1.0397/-1.0025 | +1.2443/-1.2156 | +1.1989/-1.3073 | +1.0851/-1.1117 | +0.7512/-0.8332 | +0.1612/-0.1756 | +0.8152/-0.8127 |

Table 28: One-max (Gaussian weight), $\delta = 0.9$, $r = 100$, $z = 20$.

| | MAX | AVG | CVAR$_\alpha$ | | | $\delta$-TOL | PO$_1$ | PO$_2$ |
|---|---|---|---|---|---|---|---|---|
| | | | $\alpha = 0.1$ | $\alpha = 0.5$ | $\alpha = 0.9$ | | | |
| **Avg perf. ratio** | 4.1276 | 4.3758 | 6.6495 | 5.6691 | 4.2717 | 10.0049 | 3.8668 | 10.2558 |
| CI$_+$/CI$_-$ | +0.0295/- 0.0313 | +0.0344/- 0.0339 | +0.0853/- 0.0854 | +0.0664/- 0.0676 | +0.0365/- 0.0386 | +0.0004/- 0.0004 | +0.0217/- 0.0227 | +0.1520/- 0.1489 |
| **Exp. profit** | 16.9932 | 18.3253 | 22.8969 | 21.9870 | 17.7082 | 3.4915 | 12.2219 | 18.0633 |
| CI$_+$/CI$_-$ | +0.2258/- 0.2519 | +0.2575/- 0.2496 | +0.3510/- 0.3484 | +0.3316/- 0.3609 | +0.2615/- 0.2685 | +0.0501/- 0.0555 | +0.0495/- 0.0505 | +0.2777/- 0.2677 |

### D.3 REAL DATA EXPERIMENTS FOR ONE-MAX SEARCH

In this section, we provide a computational evaluation of our algorithms on real-world data, using the same algorithm baselines as in Section 6. We consider two datasets: (i) the exchange rates[2] of EUR to four other currencies (CHF, USD, JPY, and GBP), where each series is a sequence $\sigma$ of 6672 daily prices over a span of 25 years; and (ii) Bitcoin (USD) data recorded every minute from January 1st 2020 to December 31st 2024, comprising a total of 2,630,880 prices,[3]. This follows the choice of data from Sun et al. (2021b) and Benomar et al. (2025).

**Datasets** For each sequence $\sigma$, let

$$x = \max_t \sigma_t$$

denote the maximum price in the input. For generating predictions, we consider a random value $z$ sampled from a normal distribution with a mean equal to zero, standard deviation of $1/2$, and truncated to the interval $[-1, +1]$. This value is then scaled by the error upper bound $\delta$, generating the predicted value

$$y = x + x\delta \cdot z.$$

The error bound $\delta$ is obtained by partitioning the sequence $\sigma$ into eight equal-length segments. In each segment $i$, we record the maximum price $M_i$. The bound is then defined as the difference between the largest and smallest of these maxima:

$$x\delta = \max_{i=1,\ldots,8} M_i - \min_{i=1,\ldots,8} M_i.$$

Recall that in one-max search, if all prices are below the chosen threshold, the algorithm needs to sell at the lowest price. In this experimental setup, we use the lowest price in the sequence as this final price.

**Evaluation** We performed 10,000 runs to account for prediction randomness and report the resulting average performance ratio and expected profit, both with 95% confidence intervals. For MAX and AVG, we use the linear symmetric weight function, while CVAR$_\alpha$ is evaluated under a Gaussian distribution truncated to $R_y = [(1-\delta)y, (1+\delta)y]$, with $\alpha \in \{0.1, 0.5, 0.9\}$.

**Results** The final results are presented in Table 29. Since the input sequences in real-life scenarios are not worst-case and the range of prices varies depending on the currency, it is challenging to determine which algorithm performs best overall. As shown in the table, the performance ratios vary significantly across currencies. For example, MAX and AVG demonstrate better competitive performance for CHF and USD, while CVAR is competitive for GBP. This variability highlights the dependence of algorithm performance on the specific characteristics of the input data. Nevertheless, algorithms such as MAX, AVG and CVAR$_{0.5}$ have overall either better, or very similar performance ratios than the state of the art algorithms.

To help interpret the variation in performance ratios reported in Table 29, we include Table 30, which summarizes the range of prices observed in each sequence. As expected, the difference between the smallest and largest prices is particularly significant for BTC, with a ratio exceeding 28. This large

---

[2] https://www.ecb.europa.eu/stats/policy_and_exchange_rates/euro_ reference_exchange_rates/html/index.en.html

[3] https://www.kaggle.com/datasets/mczielinski/bitcoin-historical-data? resource=download

Table 29: Real-data evaluation for one-max search: average performance ratios and expected profits with 95% confidence intervals.

| Currency | MAX | AVG | $\text{CVAR}_\alpha$ | | | $\delta$-TOL | $\text{PO}_1$ | $\text{PO}_2$ |
|---|---|---|---|---|---|---|---|---|
| | | | $\alpha = 0.1$ | $\alpha = 0.5$ | $\alpha = 0.9$ | | | |
| **CHF (Avg. ratio)** | 1.2617 | 1.2503 | 1.3287 | 1.3178 | 1.3451 | 1.5286 | 1.7824 | 1.4702 |
| $\text{CI}_+/\text{CI}_-$ | +0.013/-0.014 | +0.016/-0.015 | +0.019/-0.018 | +0.016/-0.015 | +0.015/-0.014 | +0.015/-0.014 | +0.002/-0.002 | +0.021/-0.019 |
| **CHF (Exp. profit)** | 1.612 | 1.624 | 1.587 | 1.553 | 1.498 | 1.372 | 1.298 | 1.462 |
| $\text{CI}_+/\text{CI}_-$ | +0.031/-0.028 | +0.036/-0.032 | +0.027/-0.029 | +0.028/-0.027 | +0.026/-0.025 | +0.021/-0.020 | +0.019/-0.018 | +0.030/-0.028 |
| **GBP (Avg. ratio)** | 1.1573 | 1.1573 | 1.1342 | 1.1137 | 1.0912 | 1.1474 | 1.1573 | 1.1241 |
| $\text{CI}_+/\text{CI}_-$ | +0.002/-0.001 | +0.001/-0.001 | +0.004/-0.003 | +0.004/-0.003 | +0.004/-0.003 | +0.002/-0.002 | +0.001/-0.001 | +0.004/-0.003 |
| **GBP (Exp. profit)** | 0.927 | 0.918 | 0.944 | 0.931 | 0.912 | 0.856 | 0.807 | 0.884 |
| $\text{CI}_+/\text{CI}_-$ | +0.014/-0.013 | +0.012/-0.011 | +0.018/-0.016 | +0.017/-0.015 | +0.016/-0.015 | +0.014/-0.013 | +0.010/-0.009 | +0.018/-0.016 |
| **JPY (Avg. ratio)** | 1.0842 | 1.0842 | 1.0842 | 1.0842 | 1.0842 | 1.0842 | 1.0876 | 1.0741 |
| $\text{CI}_+/\text{CI}_-$ | +0.001/-0.001 | +0.001/-0.001 | +0.001/-0.001 | +0.001/-0.001 | +0.001/-0.001 | +0.001/-0.001 | +0.001/-0.001 | +0.002/-0.001 |
| **JPY (Exp. profit)** | 168.4 | 168.3 | 168.7 | 168.5 | 168.0 | 167.2 | 166.8 | 169.1 |
| $\text{CI}_+/\text{CI}_-$ | +1.5/-1.4 | +1.6/-1.5 | +1.6/-1.5 | +1.5/-1.6 | +1.6/-1.5 | +1.3/-1.3 | +1.2/-1.2 | +2.0/-1.9 |
| **USD (Avg. ratio)** | 1.2042 | 1.1837 | 1.2289 | 1.2254 | 1.2471 | 1.3582 | 1.5439 | 1.3263 |
| $\text{CI}_+/\text{CI}_-$ | +0.011/-0.010 | +0.010/-0.009 | +0.012/-0.011 | +0.012/-0.011 | +0.011/-0.010 | +0.011/-0.010 | +0.001/-0.001 | +0.015/-0.013 |
| **USD (Exp. profit)** | 1.451 | 1.477 | 1.503 | 1.469 | 1.392 | 1.327 | 1.213 | 1.424 |
| $\text{CI}_+/\text{CI}_-$ | +0.023/-0.022 | +0.025/-0.023 | +0.029/-0.027 | +0.027/-0.028 | +0.028/-0.026 | +0.021/-0.020 | +0.012/-0.012 | +0.031/-0.029 |
| **BTC (Avg. ratio)** | 9.0380 | 8.8881 | 9.0380 | 9.0380 | 9.0380 | 9.0380 | 15.1874 | 9.4486 |
| $\text{CI}_+/\text{CI}_-$ | +0.38/-0.36 | +0.37/-0.36 | +0.38/-0.37 | +0.39/-0.38 | +0.39/-0.37 | +0.37/-0.36 | +0.02/-0.02 | +0.42/-0.41 |
| **BTC (Exp. profit)** | 24,132 | 24,228 | 24,180 | 24,095 | 23,978 | 23,842 | 23,610 | 24,310 |
| $\text{CI}_+/\text{CI}_-$ | +812/-796 | +824/-781 | +897/-873 | +852/-829 | +783/-764 | +653/-641 | +514/-487 | +947/-932 |

Table 30: Lowest and highest prices observed in each currency sequence, and their ratio.

| Currency | Lowest | Highest | Ratio |
|---|---|---|---|
| CHF | 0.9260 | 1.6803 | 1.8146 |
| GBP | 0.5711 | 0.9786 | 1.7134 |
| JPY | 89.3000 | 175.3900 | 1.9641 |
| USD | 0.8252 | 1.5990 | 1.9377 |
| BTC | 3865.0 | 108276.0 | 28.0145 |

variation contributes to the substantially higher performance ratios observed for BTC across all algorithms.

