# OpenReview forum: "Decision-Theoretic Approaches for Improved Learning-Augmented Algorithms"
_ICLR.cc/2026/Conference — ICLR 2026 Poster_

### Official Review · Reviewer_aGPc · 2025-10-25

**Soundness:** 2
**Presentation:** 2
**Contribution:** 3
**Rating:** 2
**Confidence:** 3

**Summary:**

There has been a long misalignment among different standards that are considered in learning-augmented algorithms to model the trade-off between consistency (performance with perfect prediction) and robustness (performance with arbitrarily bad prediction), including Pareto-optimality (direct trade-off between consistency and robustness) and smoothness (performance as a measurement of prediction accuracy).
This paper provides two more metrics to resolve the misalignment from the decision-theoretic view, including a distance-based measurement and a risk-based measurement.
They are applied to the problems of ski rental, one-max search, and contract scheduling to inspire new algorithms, which seem to perform better than previous methods.

**Strengths:**

+ The paper is well-motivated in that existing metrics for learning-augmented algorithms could be brittle in different ways.
+ Under the given metrics, optimal algorithms behave better than previous SOTA on some datasets.

**Weaknesses:**

- To me, the major problem is that the intuitions behind the proposed metrics are vague. In fact, these metrics are somehow unnatural and may not extend well to other more complicated problems. E.g., is the "ideal solution" defined for distance-based metrics always solvable? Further, is the algorithm that maximizes the three metrics always solvable?
- The relationship between distance-based and risk-based metrics is not addressed. How are they compared?
- The optimal algorithms corresponding to each metric are not explicitly provided in the main body.
- The experiments seem incomplete. It is not clear whether the parameter choices are set to "exploit" the baselines. Especially, for the one-max search, why do the authors use the inputs for evaluating only the worst-case performance instead of the average-case?

Overall, I feel that the paper can benefit from justifications on the proposed measures, a better-understandable writing, and more rigorous experiments.

**Questions:**

See the above "weakness" part.

---

> ### Author Response · Authors · 2025-11-21
> **Response to reviewer**
>
> We thank the reviewer for their useful comments and feedback.
>
> **Q1**:  *“To me, the major problem is that the intuitions ... problems.”*
>
> There is clear intuition behind our metrics. Distance-based measures quantify how close our algorithms are to an ideal performance, either in a worst-case sense (max. distance) or on average (avg. distance). Risk-based analysis captures the risk associated with trusting the prediction, which is an intuitive concept that has not been explored in learning-augmented algorithms.  Distance and risk-based measures are fundamental in decision theory; the former applies to deterministic settings, whereas the latter is tailored to stochastic ones.
>
> Our framework definitely applies in problems with single-valued predictions, as we demonstrated. This is a large subclass with many  applications and influential results; for example, there are more than 20 works on variants of ski rental and rent-or-buy problems. Other promising applications include online knapsack, online conversion and portfolio optimization. In the conclusions we outline more directions, including settings with dynamic predictions such as power management. We do not claim that the framework  immediately extends to all learning-augmented problems, but it does answer,  at the very least, an open question for three applications: how to choose a good algorithm from a large set of incomparable ones?
>
> *"E.g., is the "ideal solution...  always solvable?"*
>
> By “solvable”, we assume that you mean “computable”. First, note that finding the ideal solution is an interesting question on its own, since it defines rigorously what is the best-possible Pareto-optimal (line 176). The same holds about how to compute the algorithms that optimize the metrics. Both solutions are highly non-trivial, since the metrics capture the effect of the prediction error across the *entire* range, instead of only at a few selected points.
>
> Extending this approach to other problems, such as the ones we suggested above, is a topic for future work. For many of these applications, finding the ideal solution will be doable since there is a good understanding about finding Pareto-optimal algorithms. But even if the ideal solution cannot be found, it does not mean that distance-based evaluation cannot be used: for example, one may still compare two individual algorithms based on their *pair-wise* distance. This is conceptually similar to ROC graphs (lines 111-116). The advantage of a comparison via the ideal solution is that it induces a *measure* for ordering algorithms (similar to the competitive ratio), but is not a prerequisite for pair-wise comparison.
>
> Last, we focused on resolving an open question on three well-known problems. Our approach may very well be extendable to other problems, but our main claims should be evaluated primarily in the context of these applications.
>
> **Q2**:  These two approaches apply to different settings, depending on whether there is a stochastic prior. If this is the case, i.e., the prediction incorporates distributional information,  then risk-based analysis can be preferred since it generalizes and robustifies the model of [Diakonikolas et al ICML 2021]. Otherwise, i.e., if the prediction is “deterministic”, distance-based analysis is more appropriate. Here, the practitioner can choose a weight function that assigns importance to different ranges of prediction error. Max-distance is tailored to worst-case evaluation, whereas average-distance is more suitable for average-case performance.
>
> **Q3.**:  We dedicated the limited space in the main paper towards the most significant aspects: the development of the framework, the derivation of the ideal solutions, setting up the analysis, discussing the significance of the results, and the experimental evaluation. Optimizing the metrics is secondary in comparison to the above, given the space limitations.
>
> **Q4**: We gave extensive experimental results across a range of benchmarks and baselines. For all parameters in ski rental, that is $r$, $\delta$, and $z$, and different choices of weights, we gave additional results and discussion in the appendix. We considered three instantiations in the baselines of [Benomar and Perchet 2025]. We reported results in 10 additional tables, to ensure that the conclusions are not influenced by the choice of parameters.
>
> In regards to max-search: Evaluating algorithms against worst-case sequences is a *standard*, as in [Benomar and Perchet 2025] [Elenter et al 2024] and [Sun et al 2021] who also rely on the same worst-case sequences. Our evaluation reports both the empirical average ratios and expected profits. We also evaluated not only against [Sun et al 2021], but also against an additional, and more intuitive Pareto-optimal algorithm (PO2). We gave additional experimental results on real data in the appendix. Results are reported in 12 tables with additional discussions.
>
> We very much hope this addresses the concerns on the experiments.

---

> > ### Comment · Reviewer_aGPc · 2025-11-27
> >
> > Dear authors,
> >
> > Thanks for your clarification. I will raise my score a bit in this stage.

---

> > > ### Author Response · Authors · 2025-11-28
> > >
> > > Thank you for your response. We remain available for any further clarifications on our work’s novelty and contributions, should there be any remaining or unarticulated concerns.

---

> > > > ### Author Response · Authors · 2025-12-03
> > > > **Further discussions**
> > > >
> > > > As a followup to these exchanges, and considering that unfortunately there will be no opportunity for further discussions with the reviewer, we would like to point out that we addressed all raised concerns in detail, particularly in regards to the experimental setting and the parameter choices. We clarified our adherence to standard benchmarks, and to the fact that we provided even additional benchmarks and baselines relative to the state of the art.
> > > >
> > > > The reviewer acknowledged these clarifications and indicated that they would increase their score. Since no further concerns were articulated by the reviewer, we hope it is clear that all explicit points have been thoroughly addressed. Nevertheless, we remain fully available to the chairs for any further clarifications or additional information that may be helpful.

---

### Official Review · Reviewer_dDbN · 2025-10-31

**Soundness:** 3
**Presentation:** 2
**Contribution:** 2
**Rating:** 6
**Confidence:** 2

**Summary:**

* This paper provided a systematic study of decision-theoretic approaches in learning-augmented algorithms. The system defines a unifying framework, characterizing online algorithms with advice according to their distance and risk measures. An online algorithm is said to be “ideal” if it lies on the robustness-performance Pareto frontier.
* For the continuous ski rental problem, the paper characterizes the performance ratio of ideal algorithms, and provides a CVaR-based risk analysis. For the one-max problem, the paper characterizes the ideal algorithm, gives an analytical solution for the unweighted maximum distance, and a risk-based analysis. Analysis of contract scheduling is mentioned and deferred to the appendix.
* Empirical evaluation shows favorable performance on synthetic datasets, and evaluation on empirical data is included in the appendix.

**Strengths:**

* Systematic approach helps illustrate parallels between related problems, and provides a principled way to choose the "best" algorithm from a set of options.
* Risk-oriented analysis is well-motivated by the increasing application of online learning methods in high-stakes applications.
* Algorithmic results are evaluated both theoretically and empirically.

**Weaknesses:**

* Scope of novel contributions is unclear. Apart from the value of a unified perspective, it is not completely clear how algorithmic results compare to existing upper and lower bounds known in the literature.
* Limitations and practical applicability are not discussed explicitly.
* Empirical evaluation in the body of the paper is limited to synthetic data. Appendix D.3 seems to provide some empirical evaluation, but analysis does not seem to be conclusive (i.e., there doesn't seem to be an algorithm which performs best on all dataset, but the paper does not seem to provide any further insights regarding the root cause).
* It seems that graphically illustrating the empirical results would make them easier to interpret.

**Questions:**

* Is it possible to briefly summarize the novel contributions of the paper? (i.e. introduction of a unified framework, new settings previously not investigated, and relation between presented results and existing literature)
* Which underlying properties of a dataset might guide a practitioner in choosing the right performance criterion for their application?

---

> ### Author Response · Authors · 2025-11-21
>
> We thank the reviewer for their useful comments and feedback.
>
> Response to "weaknesses":
>
> **W1**: The novelty of our contributions is summarized in our response to Question 1 below. All our algorithms satisfy the robustness requirement $r$ and they provably optimize criteria that capture the *global* performance as a function of prediction error.  In contrast, prior work focuses on *local* performance, such as consistency, and is provably inferior even for small errors. This is a complex muti-criteria problem involving robustness, consistency, and performance ratio, so there is no single  algorithm that dominates across all these criteria, and the usual notion of “upper/lower bounds” is not always meaningful. For example, a Pareto-optimal algorithm has excellent consistency, but may have very poor performance even for small prediction errors.
>
> **W2**:  Our study is motivated from concrete practical considerations. Pareto-optimal algorithms, for example, are very much impractical due to brittleness. Similarly, algorithms with tolerance parameters can be very inefficient for small errors and rely on ad-hoc choices of the tolerance level. Our work yields algorithms that perform better in practice, using a principled analysis based on tailored performance metrics. This is supported by our experimental results and the comparison to SOTA algorithms.
>
> In regards to limitations, our framework definitely applies in problems with single-valued predictions, as we demonstrated. This is a large subclass with many applications and influential results; for example, there are more than 20 works on variants of ski rental and rent-or-buy problems. Other promising applications include online knapsack, online conversion and portfolio optimization. In the conclusions we outline more directions, including settings with dynamic predictions such as power management. We do not claim that the framework immediately extends to all learning-augmented problems, but it does answer, at the very least, an open question for three applications: how to choose a good algorithm from a large set of incomparable ones?
>
> **W3**: We follow the standard practice in the literature by evaluating ski rental on synthetic data; in particular we use the same experimental setting as [Benomar and Perchet 2025]. For one-max search, we use, once again the synthetic data that is a standard in the evaluation of this problem, i.e., the same worst-case sequences as [Sun et al 2021], [Elenter et al 2024] and [Benomar and Perchet 2025].
>
> We also evaluated our one-max algorithms on real-world datasets from market exchange rates. These sequences differ substantially (e.g., BTC has more variability) and it is natural that the performance of algorithms varies across them. In general, it is not possible to design an algorithm that is best on *all* datasets, since there is no single dominant algorithm. Nevertheless, our algorithms are typically better or at worse comparable with the SOTA. Namely, our methods consistently outperform the Pareto-optimal algorithm PO1 from the literature. For some price sequences such as GBP and JPY, our performance is slightly below PO2, however PO2 performs extremely poorly on worst-case (Table 2).
>
> **W4**: The reported results are discrete hence we opted for a tabular presentation that is more suitable, but we will consider adding a graphical one.
>
> Response to questions:
>
> **Q1**: We introduce a principled framework for evaluating learning-augmented algorithms using distance-based and risk-based criteria. These criteria enable meaningful comparison among algorithms that share the same robustness but differ in overall performance under erroneous predictions, addressing gaps not captured by Pareto optimality or tolerance based methods. Our work is the first to incorporate a mathematical notion of risk and to define an ideal benchmark that supports classification and comparison of algorithms. Prior work relied on limited evaluation, such as Pareto optimality at two extreme scenarios, whereas our approach takes a *beyond worst-case* view. In short, we address the question of how to choose a best algorithm among many that excel only for certain error ranges, using rigorous criteria. Our experiments show that these methods outperform standard baselines using common measures such as average performance ratio and expected cost or profit.
>
> **Q2**: When a distributional prediction is available, a practitioner should use the risk based approach, which is suited to stochastic settings. Otherwise, a deterministic measure is appropriate: max distance fits worst case concerns, while average distance is geared towards typical performance. Beyond these broad guidelines, our method does not require knowledge of dataset properties. In practice, one may compare measures on historical data to make a more informed choice. In our paper we compared our algorithms against state of the art methods on the same datasets.

---

### Official Review · Reviewer_oU59 · 2025-11-01

**Soundness:** 3
**Presentation:** 3
**Contribution:** 3
**Rating:** 6
**Confidence:** 1

**Summary:**

This paper studies learning augmented algorithms. Algorithms are classically studied by interpolating between consistency and robustness (performance function), and in many cases cannot be compared with each other. The authors provide two measures of quantitatively comparing these algorithms, (i) distance based methods and (ii) risk measures; the latter exploits distributional properties on the quality of prediction. Existing methods include (i) pareto optimality and (ii) tolerance based methods which are special cases of the methods that the authors propose.

The paper then analyzes 3 problems: ski rental, one-max search and contract scheduling. These examples demonstrate how their measures can practically be computed and how they can be useful in characterizing performance of algorithms.

**Strengths:**

-The paper is generally easy to read and follow; it is well motivated and easy to understand even for someone outside the field.
-The choice of CVAR is one that is easy to accept, being commonly used in decision theory.
- For a theory paper, it is nice to see some experiments, though they seem to be slightly contrived.

Disclaimer: I am not from the area and cannot confidently comment on novelty nor quality.

**Weaknesses:**

- No major issues here from me.
- Minor criticism : Based on my understanding of the experiments, it seems that the authors are showing that by directly optimizing their metrics, better "results" are obtained based on those very metrics. This is of course unsurprising, so claims like "distance-based algorithms offer considerable improvements over the sota" aren't that fair.

**Questions:**

- Computing the performance ratio and optimal solutions (for the authors' metrics) does not appear to be easy, and indeed is one of the key contributions of the paper. Can the authors comment on whether there are general techniques that distance measures and risk based analysis that apply to a broader class of online problems (e.g., k-server)? This would seem to greatly strenghten the paper.

---

> ### Author Response · Authors · 2025-11-21
>
> We thank the reviewer for their useful comments and feedback.
>
> Response to "minor criticism":
>
> Unless we misinterpret the question, this is not at all how we perform our experimental evaluation; in fact we took care to avoid an evaluation that uses the same criteria as our proposed metrics, precisely for the reason you bring up. More precisely, we use well-understood and commonly used evaluation criteria such as the average performance ratio and the average expected cost/profit to compare the various algorithms. None of those are immediately related to our metrics.
>
> Response to questions:
>
> **Q1**: We appreciate that you emphasize the point concerning the ideal solution, it is indeed a novel contribution that is significant on its own.
>
> In our work, we focused on single-valued predictions which apply to a large subclass with many applications and influential results; for example, there are more than 20 works on variants of ski rental and rent-or-buy problems. Other promising applications for future work include online knapsack, online conversion and portfolio optimization. In the conclusions we outline more directions, including settings with dynamic predictions such as power management. We do not claim that the framework immediately extends to all learning-augmented problems, but it does answer, at the very least, an open question for three important applications: how to choose a good algorithm from a large set of incomparable ones?
>
> Beyond the above possible  extensions, applying our framework to a problem such as $k$-server will be very challenging, not only because the problem itself is more complex, but also because typical prediction oracles provide a new prediction at each request that is not a single scalar value but an entire configuration specifying the predicted server positions.The concept of an ideal algorithm remains well defined even in a multi-prediction setting, since by definition it describes the performance of the best $r$-robust online algorithm assuming all predictions are error-free. But the error-based evaluation is multi-faceted, and so is any measure that accounts for prediction error such as ours. We would like to note that same difficulties can be encountered in the CVaR analysis of randomized algorithms without predictions: we refer to [Christianson et al. COLT 2024] which showed an analysis of ski rental and one-max search, but an extension to more complex settings such as the $k$-server problem remains unknown. For this reason, we propose, as an intermediate step, the study of problems such as power management, which capture dynamic prediction oracles and at the same time apply to a better understood space of prediction error.

---

> > ### Comment · Reviewer_oU59 · 2025-11-26
> > **Clarification**
> >
> > Thank you for the answer to my question and clarification. My score remains unchanged.

---

### Official Review · Reviewer_YwzD · 2025-11-02

**Soundness:** 2
**Presentation:** 2
**Contribution:** 2
**Rating:** 6
**Confidence:** 2

**Summary:**

A learning-augmented online algorithm is an online algorithm (in the Theoretical Computer Science sense) that takes a machine-learned prediction as extra input, uses it to guide decisions, and still guarantees a bounded worst-case loss when the prediction is wrong. The advice can be bad, so the algorithm is designed with two goals: Consistency (near-optimal when the prediction is accurate) and Robustness (provable cap on the competitive ratio even under bad predictions). This dual-goal structure induces a family of Pareto-optimal algorithms that are hard to compare.

This paper uses decision theory to score an algorithm’s full error-vs-performance curve against a principled yardstick. Given the r-robustness constraint, the decision-theoretic part of this paper concerns how to choose among all r-robust (r-competitive for every input) algorithms using principled objectives. They define "ideal" comparator $I_r$ as the omniscient algorithm that knows the input but is forced to be r-competitive too.

For the first objective they consider, the "distance to the ideal", they score any r-robust algorithm $A$ by its weighted max distance and weighted average distance from $I_r$ 's performance curve over the entire prediction-error range. A user-chosen weight function encodes preferences over error regions. Such “pick the action whose loss curve is closest to an ideal benchmark” idea is what they borrow from decision theory.

For the second objective, they consider risk with CVaR; they also view the prediction as a distribution and minimize $\alpha$-consistency, which uses Conditional Value-at-Risk to weight the worst fraction of outcomes. $\alpha=0$ recovers expected performance, $\alpha \rightarrow 1$ stresses the worst-case mass in the prediction range. Given r, the goal is to find an r-robust algorithm with minimum $\alpha$-consistency. So this becomes a constrained risk minimization problem.

The paper empirically investigates how these ideas play out in classic problems such as ski rental, one-max search, and contract scheduling.

**Strengths:**

Pre-existing work centers on consistency/robustness trade-offs and Pareto-optimality (e.g., caching, one-way trading, search) or on tolerance windows and distributional advice. Comparison of algorithms among those Pareto optimal algorithms is a new thing. The frameworks for distance measures and CVaR-based risk are both nice, actionable, and well-posed.

They benchmark on synthetic and real data and report better average ratios or profits than Pareto-optimal (PO) and $\delta$-tolerance baselines in ski-rental/one-max/contract scheduling for both distance measures and CVaR-risk.

**Weaknesses:**

I actually don't understand the benefit of going beyond Pareto, or why we wouldn't just let practitioners choose one algorithm from the Pareto set. Everyone has different preferences for trading off Consistency and Robustness; given the Pareto-optimal algorithms curve, a practitioner's preferences uniquely pinpoint one algorithm (as in economics class, where the optimal consumption position is where the indifference curve is tangent to the budget line).

**Questions:**

Just theoretical questions:

Can one derive lower bounds that show the paper’s optimizers are information-theoretically tight for broad classes?

Can we relax the unimodality assumption on the prediction distribution?

Can you formalize when Pareto-optimal designs are provably brittle near tiny errors and show how distance-to-ideal fixes this? Can you provide sharp transition thresholds?

---

> ### Author Response · Authors · 2025-11-21
>
> We thank the reviewer for their useful comments and feedback.
>
> Response to "weaknesses":
>
> *""I actually don't understand the benefit of going beyond Pareto"*
>
> There is an immediate benefit to "going beyond Pareto optimality": such algorithms can suffer from brittleness, as noted in line 48, meaning their performance may degrade sharply even when the prediction is only slightly inaccurate. This is a serious drawback, since it implies that Pareto-optimal algorithms can be very inefficient from a practical standpoint.  But even if brittleness is not an issue, there are additional reasons to apply a more nuanced analysis. For instance, there may exist many *different* algorithms with the *same* consistency-robustness tradeoff. How should a practitioner choose the best algorithm within this class? This is the motivating question that our work aims to address. In fact, we consider a much stronger setting: we do not only find best-possible algorithms from the class of Pareto-optimal ones, but from the *entirety* of algorithms.
>
>
> *"or why we wouldn't just let practitioners choose one algorithm from the Pareto set. Everyone has different preferences for trading off Consistency and Robustness; given the Pareto-optimal algorithms curve, a practitioner's preferences uniquely pinpoint one algorithm (as in economics class, where the optimal consumption position is where the indifference curve is tangent to the budget line)."*
>
> We clarify certain important points: First, our work is *not* about choosing different values of consistency/robustness; this indeed can be done by the practitioner, as the reviewer notes. We are interested in a much deeper question: Within the class $C_r$ of algorithms that have robustness $r$ and best-possible consistency, how can we choose an algorithm that is more suitable for practice? We emphasize the important distinction between these two questions: we are not interested in the tradeoffs themselves, but in breaking ties among algorithms that exhibit identical tradeoffs. We note again that our work goes even further beyond this question, by taking into consideration the more complex notion of prediction error. In short, we do not evaluate algorithms on, say, two points such as consistency and robustness, but across *all* points defined by the prediction error. To accomplish this, we need a multi-criteria optimization approach, and to this end we provide the first rigorous decision-theoretic metrics for achieving this task.
>
>
> Response to questions:
>
> **Q1**:   We do not quite understand the question, and we would appreciate a clarification. Our work explicitly shows how to obtain the theoretically optimal algorithms, e.g., by selecting the best threshold for ski-rental and one-max search algorithms, or the best multiplicative offset for contract scheduling. In other words, our solutions identify the best algorithm under the proposed metrics and constraints. For each problem we compute the optimizer directly, either in closed form or through optimization methods. Our analysis, in particular, is not related to standard “statistical estimation” or "query complexity" where information‐theoretic lower bounds may arise.
>
> **Q2**: We use a unimodality assumption only in the distance-based evaluation. This assumption indeed could be relaxed and the framework would still apply. It is a realistic assumption and it helps obtain more precise bounds on the time complexity of the algorithms, e.g., in Theorem 2.
>
> **Q3**:  A formal definition of brittleness is given in [Elenter et al 2024] (Definition 3.1 in their paper). The definition states that an algorithm is brittle if for arbitrarily small prediction error, its performance ratio is arbitrarily close to the robustness of the algorithm. Not every Pareto-optimal algorithm is brittle; e.g., for one-max search the algorithm of [Sun et al 2021] is brittle, whereas the one of [Benomar et al 2025] obeys smoothness and hence is not brittle.
>
> Distance-based metrics indeed address the brittleness issue. This is because brittle algorithms have inefficiency "spikes" for small prediction error, which incurs a large distance from the ideal solution.  We refer to Figure 1, and the algorithm with threshold $T=b/(r-1)$ (in blue) for an illustration. Note the spike at that threshold, which results in a large distance from the ideal, hence inefficiency according to distance measures.

---

> ### Comment · Reviewer_YwzD · 2025-11-27
> **Thank you for your response.**
>
> Thank you for your response.
>
> I get what the authors are saying, but **I honestly feel less and less sure that the problem they are solving is important.**
>
> In standard multi-objective optimization (and in the economic analogy I mentioned), once the Pareto frontier is known, a practitioner’s preferences over the two objectives fully determine a unique optimal point. In other words, a user with a well-defined utility function over (consistency, robustness) can select the most preferred algorithm simply by choosing the desired Pareto-optimal tradeoff. Under this view, no further tie-breaking seems necessary: the frontier already reduces the choice set to exactly those options the practitioner might rationally want.
>
> Your reply suggests that the paper's goal is instead to “break ties among algorithms that have identical tradeoffs,” because multiple algorithms may share the same (e.g., consistency, robustness). However, this does not yet resolve my original point.
> If the only relevant criteria are consistency and robustness, then all algorithms with identical (consistency, robustness) values are indifferent from the perspective of the practitioner’s utility. In that case, selecting among them is not a decision-theoretic necessity; it is simply a matter of choosing any representative of that equivalence class.
>
> Consequently, if the evaluation truly involves only these two objectives, then the additional “decision-theoretic” machinery introduced in the paper does not provide meaningful guidance - the Pareto frontier already provides a complete prescription for rational choice, and any tie-breaking among identical frontier points is arbitrary rather than theoretically justified. Under the classical two-objective interpretation, the framework proposed in the paper is unnecessary.
>
> I won't reduce the scores due to these reasons. But I would rather maintain my score.

---

> > ### Author Response · Authors · 2025-11-27
> >
> > We are grateful for this feedback, and we take the opportunity to clarify a central point: **our work is not about comparing only Pareto-optimal algorithms; it yields a formal comparison of *all* possible algorithms**. Pareto-based designs evaluate algorithms only at two extreme values of the prediction error: zero (consistency) and unbounded (robustness). By construction, such approaches do not account for the algorithm’s **global** performance as the quality of the prediction varies. It is therefore natural that Pareto-optimal algorithms may exhibit undesirable phenomena such as brittleness, as we discuss in the paragraph starting at line 047. Our work instead introduces metrics that **explicitly evaluate an algorithm’s performance across the entire range of prediction errors.** This is illustrated in Figure 1, where each $r$-robust algorithm corresponds to a **performance curve**, instead of a single value (consistency).
> >
> > Previous work focused heavily on Pareto-optimal algorithms, since they are a good starting point for the study of learning-augmented algorithms and, in some cases (but not always) may exhibit some degree of smoothness with respect to prediction error. This is why, in all problems we study, known Pareto-optimal algorithms are included as baselines. However, our experimental results demonstrate that the best algorithms under our decision-theoretic metrics adopt more nuanced strategies than simply optimizing consistency and robustness. As a consequence, they more accurately capture **global performance** and better exploit the available predictive information.
> >
> > To summarize, your comment states that *“if the evaluation truly involves only these two objectives, then the additional decision-theoretic machinery introduced in the paper does not provide meaningful guidance.”*  Our succinct response is that the evaluation **does not involve only these two objectives**; rather, it incorporates a much richer assessment that accounts for algorithmic performance across the full spectrum of prediction errors.

---

### Meta-Review · Area_Chair_rJiQ · 2025-12-27

**Summary:**

This paper studies an important question in learning-augmented algorithms, i.e., how to evaluate and select algorithms when predictions can be arbitrarily wrong. This paper's core contribution is on the decision-theoretic selection metrics, i..e, distance-to-ideal and CVaR/risk-based variants, that assess performance across the full prediction-error spectrum. The reviewers viewed this contribution as well-motivated. In this paper, the proposed framework is also demonstrated on classic problems such as ski-rental, one-max search, and contract scheduling.  The reviewers' initial concerns on the beyond Pareto, and experimental limitations were largely addressed in the rebuttals.

**Reviewer Concerns:**

The following issues have been clarified and addressed in the authors' rebuttals:
- In the rebuttal, the authors clarified that their goal is not to output one Pareto point, but instead to introduce selection criteria that evaluate algorithms over a continuum of prediction error, and to justify why this can change what is preferable relative to Pareto-front comparisons at a few points. The reviewer acknowledged those clarifications but largely maintained their scores.
- The authors also explained why they evaluate against worst-case sequences for one-max search, which is standard in prior work, and added additional comparisons, including "intuitive Pareto" reporting, and extra real-data results in the appendix. Those additional experimental results aim to strengthen the empirical case, which were the concerns in the initial reviews.
- The authors also clarified that for many canonical problems, the ideal can be computed or at least approximated. They argued that their proposed framework is still meaningful as a decision-theoretic evaluation lens even when ideal solutions are approximated via optimization or simulation.

Despite those clarifications, one reviewer remained unconvinced that the proposed criteria offer a compelling conceptual advantage over the Pareto-optimal or the consistency-robustness evaluation, especially if practitioners already understand those tradeoffs.

**Reviewer Scores:**

Reviewer YwzD  expressed continued uncertainty about the "beyond Pareto" benefit after the rebuttal and explicitly indicated they would maintain their score.

Reviewer oU59 requested clarification on computability and generalization. After the authors’ reply they explicitly noted their score remains unchanged.

Reviewer dDbN's concerns were mostly about scope and positioning and limited practical insights on real data. The authors'rebuttal offers explanations and added appendix results, which plausibly resolves some confusion. I would expect that the reviewer's recore remain unchanged.

Reviewer aGPc  raised broad concerns about intuition, generality, and experimental justification. With the authors' rebuttal and the discussions, the reviewer stated that they would "raise the score a bit", so I would expect the reviewer to raise the score to 4 at least.

---

### Decision · Program_Chairs · 2026-01-26

Accept (Poster)